# Stimuli-Responsive Principles of Supramolecular Organizations Emerging from Self-Assembling and Self-Organizable Dendrons, Dendrimers, and Dendronized Polymers

**DOI:** 10.3390/polym15081832

**Published:** 2023-04-09

**Authors:** Virgil Percec, Dipankar Sahoo, Jasper Adamson

**Affiliations:** 1Roy & Diana Vagelos Laboratories, Department of Chemistry, University of Pennsylvania, Philadelphia, PA 19104-6323, USAjasperad@sas.upenn.edu (J.A.); 2Department of Medicine, University of Pennsylvania, Philadelphia, PA 19104-6323, USA; 3Chemical Physics Laboratory, National Institute of Chemical Physics and Biophysics, Akadeemia tee 23, 12618 Tallinn, Estonia

**Keywords:** stimuli-responsive principles, internal-stimuli, external-stimuli, supramolecular organizations, self-assembling and self-organizable, dendrons, dendrimers, dendronized polymers

## Abstract

All activities of our daily life, of the nature surrounding us and of the entire society and its complex economic and political systems are affected by stimuli. Therefore, understanding stimuli-responsive principles in nature, biology, society, and in complex synthetic systems is fundamental to natural and life sciences. This invited Perspective attempts to organize, to the best of our knowledge, for the first time the stimuli-responsive principles of supramolecular organizations emerging from self-assembling and self-organizable dendrons, dendrimers, and dendronized polymers. Definitions of stimulus and stimuli from different fields of science are first discussed. Subsequently, we decided that supramolecular organizations of self-assembling and self-organizable dendrons, dendrimers, and dendronized polymers may fit best in the definition of stimuli from biology. After a brief historical introduction to the discovery and development of conventional and self-assembling and self-organizable dendrons, dendrimers, and dendronized polymers, a classification of stimuli-responsible principles as internal- and external-stimuli was made. Due to the enormous amount of literature on conventional dendrons, dendrimers, and dendronized polymers as well as on their self-assembling and self-organizable systems we decided to discuss stimuli-responsive principles only with examples from our laboratory. We apologize to all contributors to dendrimers and to the readers of this Perspective for this space-limited decision. Even after this decision, restrictions to a limited number of examples were required. In spite of this, we expect that this Perspective will provide a new way of thinking about stimuli in all fields of self-organized complex soft matter.

## 1. Introduction

This invited Perspective will start with the discussion of relevant definitions of stimulus and stimuli as available in classic dictionaries and in various fields of science. Subsequently, the selection of the topic and of the title of this article will be justified. 

The American Heritage College Dictionary defines stimulus as “*something causing or regarded as causing a response*” [1]. Oxford Concise Science Dictionary refers to stimulus as being “*any change in the external or internal environment of an organism that provokes a physiological or behavioral response in the organism*” [2]. Academic Press Dictionary of Science and Technology considers stimulus “*a condition or change in the environment that produces a behavioral response*” [3]. In biology stimulus is “*something that produces a reaction in a plant, animal or person*.” The biological definition of stimulus is more complex and complete and provided the inspiration for the organization of this Perspective. “*A stimulus is anything that can trigger a physical or behavioral change. Stimuli can be external or internal. An example of external stimuli is your body responding to a medicine. An example of Internal stimuli is your vital sign changing due to a change in the body.*” Since supramolecular organizations or systems hierarchically emerging from self-assembling and self-organizable dendrons, dendrimers, and dendronized polymers resemble the complexity of the biological systems we will rely on the biological definition of stimulus since it differentiates between 3-dimensional (3D) supramolecular assemblies and amorphous or liquid dendrimers, polymers, and other forms of complex self-organized condensed matter. Therefore, a brief discussion of the historical developments of conventional dendrons, dendrimers, and dendronized polymers will be compared with the evolution of supramolecular organizations assembled from self-assembling and self-organizable dendrons, dendrimers, and dendronized polymers. Synthetic and structural analysis methodologies from both fields will be summarized before a comprehensive classification of internal and external stimuli in supramolecular dendrimers and in their periodic and quasiperiodic assemblies, accompanied by selected examples, will be provided. We expect that this Perspective will impact the way of thinking about stimuli in other classes of complex self-organized supramolecular systems.

## 2. The Discovery of Dendrimers and Their Development

Dendrimers were discovered independently in four different laboratories. In 1978, the laboratory of Fritz Voegtle from the University of Bonn [4] reported the bis-cyanoethylation of primary aliphatic amines and of bifunctional secondary amines followed by reductions of the resulting cyano groups and subsequent iteration of the cyanoethylation to produce two generations of what they named at that time “cascade” molecules. In a series of U.S. patents starting in 1981 Denkewalter, Kolc, and Lukasavage of Allied Corporation [5] elaborated the first examples of globular poly(lysine) that were demonstrated by Aharoni from the same company to be globular and monodisperse [6]. In 1985, Tomalia, Baker, Dewald, Hall, Kallos, Martin, Roeck, Ryder, and Smith from Dow Chemical Company reported the synthesis and characterization of a new class of polymers named starburst-dendritic macromolecules [7]. Seven generations of starburst-dendritic macromolecules were prepared by the iterative Michael addition of methyl acrylate to ethylenediamine or ammonia followed by amidation with excess ethylene diamine and reiteration of the same process. This comprehensive publication established the name dendrimer and pioneered the field of dendrimers. Also in 1985, Newkome, Yao, Baker, and Gupta reported a new “cascade” molecule that provided a synthetic approach to unimolecular micelles. This molecule was named Arborol [8]. All four laboratories provided excellent synthesis methodologies, identified side reactions leading to molecular imperfection(s) in the structure, and performed careful characterization of the resulting dendrimers.

Influential review articles and books on dendrimers were published by Tomalia, Voegtle, and Newkome laboratories and by many other laboratories that joined the field soon after [9,10,11,12,13,14,15,16,17,18,19,20,21,22,23,24,25,26,27,28]. Numerous applications emerged from the field of classic dendrimers. Most of these dendrimers and dendronized polymers are liquid or amorphous solids and even when they crystallize, their structure was not solved at the molecular level. They were discussed in numerous review articles and books and will not be presented again in this Perspective.

## 3. The Discovery of Self-Assembling and Self-Organizable Dendrons, Dendrimers, and Dendronized Polymers

In an attempt to prepare a biaxial nematic liquid crystal from a molecule combining a half-disc with a rod-like fragments, during the late 1980s, Percec and his graduate student Heck discovered both self-assembling dendrons and self-organizable dendronized polymers [29,30,31,32,33,34,35,36,37,38,39,40,41,42,43,44,45,46,47,48,49,50,51,52,53,54,55,56,57,58,59,60]. The half-disc fragment employed in these experiments was a first-generation self-assembling dendron found to self-organize in the absence and presence of a large diversity of chemical fragments attached at its apex including rod-like, crown-ethers, oligooxyethylenes, their metal salt complexes, metal carboxylates, and polymers. All these combinations were discovered to self-assemble and self-organize into supramolecular helical columns resembling Tobacco Mosaic Virus (TMV). Figure 1 outlines the divergent and convergent methodologies for the synthesis of self-assembling dendrons and dendrimers and the simplest molecular principles that were employed to transform non-self-assembling dendrons and dendrimers into the corresponding self-assembling building blocks.

## 4. Methodologies Employed in the Synthesis of Self-Assembling and Self-Organizable Dendrons, Dendrimers, and Dendronized Polymers

These methods will be only briefly enumerated in case the reader would like to become familiar with organic, supramolecular, and polymer synthesis methodologies, some of them specifically designed for this process. Cu(I)-catalyzed azide-alkyne, thio-bromo click, and TERMINI double click reactions [61,62,63,64,65,66,67,68,69,70,71,72,73,74] and Ni-catalyzed borylation and cross-coupling [75,76,77,78,79,80,81,82,83,84,85,86] are only a few less conventional organic reactions employed to elaborate new and efficient iterative and modular-orthogonal strategies for the synthesis of self-assembling and self-organizable dendrons, dendrimers, and dendronized polymers. Poly(methylsiloxane)s with well-defined molar mass and narrow polydispersity were prepared by living ring-opening polymerization [87], poly(vinyl ether)s by living cationic polymerization of vinyl ethers [88,89], substituted polyethylene imines by living cationic ring opening polymerization of oxazolines [90], polyacrylate, and polymethacrylate by group transfer polymerization [91] and SET-LRP [71,72,73,74,91,92,93,94], helical poly(phenyl and aryl acetylene)s with controlled stereochemistry by living polymerization [95,96,97,98,99,100,101], by living metathesis ring opening [52] and living anionic polymerizations [51], all elaborated for the conventional synthesis of side-chain liquid crystal polymers and of other complex systems [102,103,104] were employed in these experiments. The story of this discovery was published by invitation including in several review articles [98,102,103,104,105,106,107,108,109,110,111,112]. 

**Figure 1 polymers-15-01832-f001:**
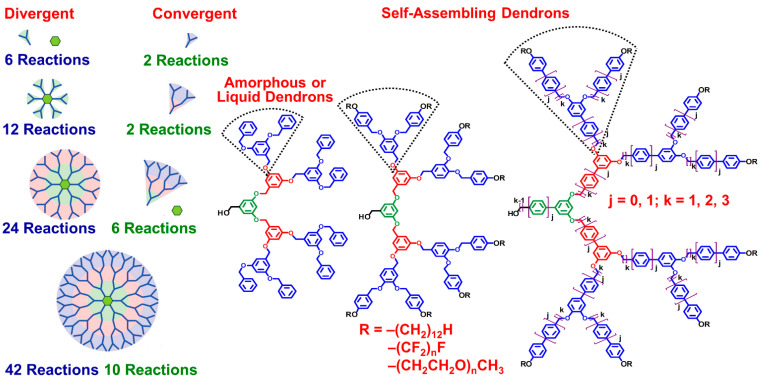
Divergent and convergent methodologies for the synthesis of dendrimers and dendrons displaying the number of reactions (left side); Examples of amorphous or liquid dendrons and self-assembling dendrons (right side) [60]. The parts of the Figure were adapted from [111] combined and modified. Copyright © 2009, American Chemical Society.

## 5. Discover, Elucidate the Mechanism, and Predict Primary Structure by Generational, Deconstruction and Other Synthetic Methodologies Combined with Structural and Retrostructural Analysis

Figure 2 outlines the structural and retrostructural analysis of supramolecular assemblies performed by a combination of differential scanning calorimetry (DSC), thermal optical polarized microscopy (TOPM), powder and oriented fiber X-ray analysis (XRD) combined with electron density maps, transmission electron microscopy (TEM), and other methods [60,112]. This process allows for determining the secondary structure of the self-assembling dendrons, dendrimers, or dendronized polymers. Generational libraries of self-assembling dendrons constructed by the convergent process illustrated on the left side of Figure 1 were employed for these investigations. Examples of representative secondary structures are shown at the top of Figure 2. Their secondary structures are in the middle row of the same Figure 2, while tertiary structures are in the bottom row. Various columnar assemblies self-organized from helical columns [40,41,42,51,52,55,113] are shown on the left side, whereas periodic and quasiperiodic arrays self-organized from spherical helices including liquid-quasicrystal [114,115]. We feel that at this time providing brief definitions of self-assembly and self-organization could be helpful. Self-assembly refers to the aggregation of a supramolecular object, whereas self-organization refers to the process via which a periodic or quasiperiodic array generated from supramolecular objects is formed [71,101].

Frank–Kasper A15 [58,116,117,118,119,120,121,122,123,124,125,126,127,128,129,130,131] and σ [132] as well as BCC [133,134] are illustrated on the right side of Figure 2. A few examples of dendronized polymers indicate that the polymer backbone will adopt a helical conformation confined to the center of the helical columns or a random coiled or helical conformation in the case of spherical helices (last two models in the right side of Figure 2).

Elucidation of the mechanism of self-assembly and self-organization provides access to the prediction of the primary structure that will undergo a specific mechanism of self-organization with about 85% predictability [131]. Once this level of predictability is accomplished, the probability to discover new primary structures that will generate new secondary, tertiary, and quaternary structures by the generational methodology, decreases. Subsequently, new synthetic approaches to self-assembling dendrons were developed. The deconstructions strategy [135] to discovery was developed to discover new primary structures of self-assembling dendrons. The deconstruction methodology is illustrated in Figure 3 [135]. 

A previously synthesized and characterized self-assembling dendron or a structure that was never synthesized is cleaved on paper, and the imaginary new dendrons whose primary structure would not have been predicted by the generational methodology is synthesized. Their structural and retrostructural analyses is performed as in the case of the generational libraries. The deconstruction strategy to discovery is illustrated in Figure 3 [135].

The two primary structures highlighted in yellow indicate some of the most important discoveries generated by this strategy. One from the left side of Figure 3 provided a molecular switch to be discussed in more detail in Figure 4, whereas the one from the right side of the same figure shows a 1.7 × 10^6^ gmol^−1^ monodisperse bilayered spherical assembly generated from 933 self-assembling dendrons that is in the range of the size of the ribosome [135]. This structure had a diameter of 171.9 Å.

Figure 4 outlines a diversity of supramolecular assemblies generated from the primary structures designed by the deconstruction strategy are outlined in Figure 4. The third structure from the left side of this Figure demonstrates a new self-assembly mechanism that provides access to a molecular switch. At low temperature, a columnar hexagonal periodic array is assembled. At a higher temperature, the columns transform into ondulated columns, and the periodic array transforms from a 2D lattice with *a* = 65.3 Å) into a 3D superlattice (with *a* = 175.3 Å). The ondulated columns double the size of the lattice when the superlattice is generated (ACIE). The third approach to discovery via libraries was produced by changing the architecture of self-assembling dendrons from being constructed via *sp*^2^–*sp*^3^ bonds to *sp*^2^–*sp*^2^ bonds [85,86]. This process was possible by replacing benzyl and alkyl ethers as links in the structure of self-assembling dendrons with aryl-aryl *sp*^2^–*sp*^2^ bonds. This was possible by employing two new chemistries elaborated in our laboratory: Ni-catalyzed borylation with Ni-catalyzed Suzuki-like cross-coupling [75,85,86]. A large number of symmetric and non-symmetric constitutional isomeric dendrons were generated. A library of symmetric AB_2_ to AB_9_ self-assembling dendrons was generated by a single and simple alkylation with an *n*-alkyl benzyl chloride [86]. Two constitutional isomeric AB_2_, four constitutional isomeric AB_4_, two pairs of constitutional isomeric AB_6_, one AB_3,_ and one AB_9_ self-assembling dendrons were generated (Figure 5) [85,86]. Before this work was reported, AB_4_ and AB_5_ were the most complex self-assembling dendrons that were synthesized by a very large number of steps [130]. The discoveries generated with the structures from Figure 5 will be discussed in a later chapter of this Perspective.

## 6. Classification of Internal and External Stimuli in Supramolecular Dendrimers

Based on the discussion from the previous subchapters, particularly from Figure 2, Figure 3, Figure 4 and Figure 5, that illustrate the role of the primary structure on the tertiary, quaternary structures of supramolecular assemblies, it is quite clear that stimuli in supramolecular dendrimers can and, we believe, it must be classified as in biological systems into internal and external stimuli. A classification based on these principles is illustrated in Figure 6. In our classification, internal stimuli refer to the role of all components of the primary structure of self-assembling dendrons and dendrimers as well as of dendronized polymers on their secondary and tertiary structures resulting via self-assembly and self-organization. External stimuli refer to the role of the environment, temperature, light, electric, and magnetic fields on the functions of the supramolecular structure and of its co-assembly with various components of the environment under a set of conditions. In the next subchapters, we will discuss with selected examples, mostly from our laboratory, how all internal and external stimuli affect the functions of supramolecular dendrimers and of the supramolecular dendronized polymers.

## 7. Internal Stimuli of Supramolecular Dendrimers. Primary Structure of Dendrons and Dendrimers

The internal stimuli of supramolecular dendrimers are provided by the primary structure of the self-assembling dendrons and dendrimers. In this chapter we will discuss with selected, and usually simple examples, how various components of the primary structure of the self-assembling dendron or dendrimers affect their supramolecular structure. The primary structure of self-assembling dendrons and dendrimers is as influential in affecting the structure of supramolecular dendrimers as the primary structure of peptides and proteins affects their secondary, tertiary, and quaternary structures.

### 7.1. Chemical Composition

We will select an example in which alkyl groups of a second generation AB_3_ dendron affect the secondary, tertiary, and quaternary structure of its supramolecular dendrimer (Figure 7). When the *n*-alkyl groups on the periphery of this dendron are perhydrogenated, the dendron adopts a conical secondary structure that self-assemble a supramolecular sphere as a tertiary structure. The supramolecular sphere will self-organize into a cubic *Pm*$\overset{-}{3}$*n* lattice known also as an A15 Frank–Kasper phase. When the *n*-alkyl groups of the same dendron are semifluorinated, the dendron changes its secondary structure from conical to crown conformation [136]. The crown conformation self-assembles into supramolecular pyramidal columns rather than spheres, and the supramolecular pyramidal columns self-organize into a periodic columnar hexagonal p6mm array. Similar or different situations were demonstrated by changing the chemical composition of the dendrons or dendrimers in numerous libraries.

### 7.2. Sequence

Figure 8a provides an example in which the chemical composition of two AB_3_ self-assembling dendrons is identical, but the sequence in which the components of the two building blocks is different [127]. A conical conformation is adopted by (4-3,4,5-4^3^)12G1-CH_2_OH while, depending on the temperature, a triangular or tapered secondary structure is adopted by (4^2^-3,4,5)12G1-CH_2_OH. Subsequently, the first self-assembling dendron assembles supramolecular spheres forming a cubic *Pm*$\overset{-}{3}$*n* or A15 Frank–Kasper phase, whereas the second dendron self-assemble, depending on temperature, a bilayer forming a smectic liquid crystal phase or a supramolecular column generating a columnar hexagonal *P*2*mm* or *P*6*mm* symmetry. Figure 8b demonstrates an additional example in which the sequence of a combination of linear *n*-nonyl and branched chiral or racemic labeled, in, this case as an *r* group attached to a perylene-bisimide provides an extraordinary acceleration for the formation of a cogwheel supramolecular helix; that is, the case of *rrr*-PBI or 999-PBI sequence is not observed at all or requires a very long annealing time at high temperatures to form it, as in the case of PBI-*rrr*. Compare the DSC traces of constitutional isomeric sequences of *r*9*r*-PBI with *rr*9-PBI and with the constitutional isomeric sequences 9*r*9-PBI with 99*r*-PBI from the bottom part of Figure 8b [137,138,139].

### 7.3. Constitutional Isomerism

The two self-assembling dendrons from Figure 8 provide also an excellent example of how identical chemical composition but different constitutional isomerism provide different supramolecular structures and functions. Additional examples of constitutional isomerism are available in Figure 5 [86]. The best examples are provided by the four AB_4_ constitutional isomers marked in yellow. IVd self-organized cubic *Pm*$\overset{-}{3}$*n* or Frank–Kasper phases regardless of the functional group available at its apex. Vd and VIId self-organize different columnar phases, whereas VIIId self-organized bilayered smectic phases.

### 7.4. Multiplicity of Branching Point

Figure 9 provides an example that demonstrates the role of the multiplicity of the branching point during the self-organization of the supramolecular assembly [86]. The self-assembling dendrons (3,4-(3,4,5)*^n^*^−1^)12Gn-X and (3,4-(3,5)*^n^*^−1^)12Gn-X were selected as models for this discussion [126]. They will be discussed for two different generations. The first self-assembling dendron shown in the left side of Figure 9 has an AB_3_ branching point, whereas the second one has an AB_2_-derived branching point. The structure on the periphery of both self-assembling dendrons is identical. The transition from the 3,4,5-trisubstituted AB_3_ benzyl ether to the 3,5-disubstituted AB_2_ benzyl ether in the inner part of the structure changes completely the mechanism of self-assembly and self-organization. In the first case, a conical secondary structure is adopted by the self-assembling dendron, whereas in the second case, a tapered secondary structure is adopted. The conical dendron self-assembles into supramolecular spheres forming *Pm*$\overset{-}{3}$*n* cubic or A15 Frank–Kasper phase. In the second case, the tapered secondary structure self-assembles a supramolecular column forming a columnar hexagonal periodic array. This dependence between the multiplicity of the branching point and the hierarchical mechanism of self-organization was observed for all libraries of self-assembling dendrons investigated in our laboratory and in other laboratories [111,112,113,114,115,116,117,118,119,120,121,122,123,124,125,126,127,128,129,130,131]. The role of the multiplicity of the branching point in non-self-assembling dendrons and dendrimers forming liquid or amorphous solids is not accessible via X-ray diffraction experiments and, therefore, is not as precisely known as in the case illustrated in Figure 9.

### 7.5. Constitutional Isomerism of the Branching Point

The constitutional isomerism of the branching point in self-assembling dendrons is as influential during the self-assembly process as the multiplicity of the branching point discussed in the previous subsection. Figure 10 provides an example that demonstrated this concept [126]. 

This discussion will be made for three different generations of the two self-assembling dendrons shown in Figure 10. The first generation located on the periphery of the self-assembling dendrons (4-3,4-(3,5)^n−1^)12Gn-X (see the left side of Figure 10) and (4-(3,4)^n^)12Gn-X (see the right-side part of Figure 10) are identical. However, the inner part of the self-assembling dendron from the left side of Figure 10 has a 3,5-disubstituted benzyl ether, while the inner part of the self-assembling dendron from the right side has a 3,4-disubstituted benzyl ether. The 3,5- and 3.4-disubstituted branching points are both AB_2_, but at the same time, they are constitutional isomers. Generation one (n = 1) of both dendrons exhibits a tapered secondary structure forming supramolecular columns. However, generations two and three undergo a completely different hierarchical mechanism of self-assembly and self-organization for the corresponding dendrons. The 3,5-disbusbstituted dendrons continue via the same mechanism of self-assembly as its first generation, that is, tapered secondary structure leading to a supramolecular column forming a columnar hexagonal periodic array. The main difference, as a function of generation number, consists of the number of tapered dendrons forming the cross-section of the supramolecular column and its diameter. As we change from the 3,5-disubstituted pattern to its constitutional isomeric 3,4-pattern the hierarchical mechanism of self-organization changes. The 3,4-constitutional isomeric pattern-based dendron adopts a conical secondary structure that forms a supramolecular sphere assembling into cubic *Pm*$\overset{-}{3}$*n* or Frank–Kasper A15 periodic arrays. This process is similar for both generations two and three except that the number of conical dendrons forming the sphere increases, and the diameter also increases as the generation number increases (right side of Figure 10) [126]. Since spherical lattices exhibit different physical properties from columnar hexagonal lattices, the constitutional isomerism of the branching point illustrated in Figure 10 is similar to the constitutional isomerism of cellulose and amylose. Both cellulose and amylose have strictly the same chemical composition but different stereochemistry at the anomeric carbon and, therefore, are constitutional isomers with completely different chain conformations, liner for cellulose and helical for amylose, different solubilities in water, and different physical properties. Therefore, Figure 10 provides an excellent example correlating constitutional isomerism of the self-assembling dendrons with their supramolecular structure and function. The role of constitutional isomerism of the branching point in amorphous and liquid dendrimers was not yet elucidated.

### 7.6. Generation Number

Figure 11 illustrates the hierarchical self-organization of the same self-assembling dendron at two different generations. (4-(3,4,5)^2^12G2-X is a second generation (top of Figure 11), whereas (4-(3,4,5)^3^12G3-X (bottom of Figure 11) is a third generation [124,125]. Change in the shape of the dendrimer as a function of generation was predicted by Tomalia, but it was difficult to demonstrate it definitively with this PAMAM dendrimer [9]. In 1998, our laboratory demonstrated, by X-ray diffraction experiments, the change in shape as a function of generation for supramolecular dendrimers [124,125]. Figure 11 provides an example that demonstrates this concept. The second-generation self-assembling (4-(3,4,5)^2^12G2-X adopts a half of a disc secondary structure that self-assembles into supramolecular columns forming columnar hexagonal periodic arrays. The third generation (4-(3,4,5)^3^12G3-X adopts a fragment equal to ^1^/_6_ of a spherical secondary structure that self-organizes spherical supramolecular dendrimers generating *Pm*$\overset{-}{3}$*n* cubic phases known also as A15 Frank–Kasper phase. This general concept was demonstrated with numerous generational libraries of self-assembling dendrons [60,112,124,125,126,127,128,129,130,131,133].

### 7.7. Covalent and Supramolecular Multiplicity of the Focal Point

Multiplicity of the focal point can be covalent or supramolecular. Covalent multiplicity of the focal point is forced by the chemical structure employed to construct the focal point with self-assembling dendrons and is limited by the chemical capabilities available to generate multiplicity. Figure 12 illustrates several examples of architectures employed frequently in our laboratory to assemble the branching point. Multiplicities of 2, 3, and 6 that are commonly employed in our laboratory are shown in Figure 12. Supramolecular multiplicity at the focal point is spontaneously determined by self-assembly and is practically unlimited as numbers. It usually matches the multiplicity number that is required to fill the space by self-assembly. We would like to point to a few examples of the supramolecular multiplicity of the focal point by inspecting Figure 8 [27]. In the supramolecular columns from the right side of Figure 8, we can see a supramolecular multiplicity of 19 and 6. While a multiplicity of 6 can be also accomplished by the covalent strategies indicated in Figure 12, no covalent mechanism is known today to accomplish a covalent multiplicity of 19. An even more striking demonstration of the supramolecular multiplicity at the focal point is observed on the left side of Figure 8. A supramolecular multiplicity of 269 is endowed by the self-assembly of the supramolecular spheres from the conical secondary structure of the dendron [131]. No covalent mechanism is known to accomplish a multiplicity of 269! Numerous examples of giant supramolecular multiplicities at the focal point can be seen by screening through libraries of self-assembling dendrons in original publications and review articles from our laboratory [58,60,61,62,63,64,65,66,67,68,69,70,71,72,73,74,75,76,77,78,79,80,81,82,83,84,85,86,87,88,89,90,91,92,93,94,95,96,97,98,99,100,101,102,103,104,105,106,107,108,109,110,111,112,113,114,115,116,117,118,119,122,124,125,126,127,128,129,130,131,132,133,134,135,136,137,138,139,140,141,142,143,144,145,146,147,148,149,150,151,152,153,154,155,156,157,158,159,160]. The trivial question is: what happens to the self-assembly process when the supramolecular multicity is replaced with a covalent multiplicity of a much lower value? The answer is very simple, a new mechanism of self-assembly and self-organization will be generated, and we will not go into any details in this Perspective on this issue.

### 7.8. Constitutional Isomerism of the Focal Point

Figure 12 presents also two covalent constitutional isomers of the focal point: 1,3,5- and 1,2,3-trisubstituted benzene based constitutional isomers [124]. In one experiment in which they were compared, they both provided the same mechanism of self-organization of a supramolecular column except that the thermal stability of the column assembled from the 1,2,3-focal point is much higher than the one assembled via the 3,4,5-based focal point [124].

### 7.9. Stereochemistry at the Focal Point and in Other Parts of Self-Assembling Dendrons and Dendrimers

We will start this discussion with the self-assembly of dendritic dipeptides elaborated in our laboratory to self-assemble mimics of the helical water channel aquaporin (AQP) [161,162,163,164,165,166,167,168,169,170]. Figure 13a illustrates the concept of the helical dendritic dipeptide and of its self-assembly into hydrophobic helical pores [161,162,163,164,165,166,167,168,169,170]. A dipeptide based on tyrosine and any other α-amino acid containing a variety of protecting groups and all possible stereochemical permutations of the two α-amino acids were attached at the focal point of a self-assembling dendron generating supramolecular columns (see top-part of Figure 13a). 

A solvophobic solvent that mimics the hydrophobic part of the cell membrane was found to select the tapered conformation of the dendrons that mediates the self-assembly of helical pores both in solution, in the cell membrane, and in the bulk state. Homochiral dendritic dipeptides self-assemble highly ordered single-handed porous dendritic dipeptide columns, heterochiral self-assemble less ordered porous dendritic dipeptides display a lower rate of crystallization, whereas racemic dendritic dipeptides assemble micellar disordered porous columns that do not crystallize and due to their very strong H-bonding cannot be deracemized even upon long time annealing. Homochiral dendritic dipeptides assemble helical pores that transport water, while their racemic structure does not. This is a very powerful example demonstrating the role of chirality and the transfer of its stereochemical information from the focal point to the crystal structure of the supramolecular assembly and of its functions. The replacement of the 3,5-disubstituted pattern of the dendritic dipeptide with a 3,4-pattern changes the helical porous assembly into a hollow spherical helix, demonstrating, again, the power of constitutional isomerism. When the strong H-bonding forming the helical assembly from dendritic dipeptide is replaced with a less strongly interacting arrangement generated by a dendronized cyclotriveratrylene (CTV) (Figure 13b), a hat-like homochiral or racemic column is generated [171]. The racemic and homochiral assemblies self-organize helical columns just like in the case of the dendritic dipeptides. A low order is observed in the racemic columnar hexagonal array of the hat-like molecules (Figure 13b). However, upon annealing this racemic structure at 60 °C for 2 h, a deracemization in the crystal state occurs, and a highly ordered columnar hexagonal crystal is generated. This is the first example of deracemization in the crystal state from the entire literature. The driving force for this deracemization is provided by the unit cell of the hexagonal crystal that contains a single column or four quarters of columns that must all exhibit the same handedness in order to provide a highly ordered hexagonal crystal array.

### 7.10. The Three-Dimenional Architecture of the Focal Point

As shown in Figure 12, covalently generated focal points can be planar, as in the case of trihydroxybenzene (THB) and hexahydroxytriphenylene-based structures, conformationally disordered as in the case of hexasubstituted tetraveratrylene (CTTV), or exhibit preferentially a crown conformation as is the case for trisubstituded or hexasubstituted cyclotriveratrylene (CTV). THB has been shown to induce predominantly a conical of tapered dendron or dendrimer secondary structure [98,118,168,172,173], whereas triphenylene focal point planar architecture favors the formation of crown-like supramolecular dendrimers [174]. Enhancing the conformational flexibility of dendronized triphenylene via diethylene glycol linkers lowers transitions of helical columnar, Frank–Kasper, and quasicrystal phases [175]. The crown conformation resulting from triphenylene is capable of promoting the self-organization of both helical pyramidal columns and spherical helices [174,175,176,177,178,179]. The conformationally disordered but flexible CTTV induces similar supramolecular structures to triphenylene except that they are more flexible and display higher dynamics of their self-organization. CTV was the model employed for the self-assembly of pyramidal columns and spherical helices [176,177,178,179,180,181,182,183,184,185,186].

## 8. Internal Stimuli of Supramolecular Dendrimers. Primary Structure of Self-Organizable Dendronized Polymers

The internal stimuli of supramolecular dendrimers self-organized from dendronized polymers is more complex than the primary structure of self-assembling dendrons and dendrimers since it takes into account the primary structure of the dendron or dendrimer, the secondary structure of the dendron or dendrimer, the degree of polymerization of the backbone of the dendronized polymer, and the stereochemistry of the polymer backbone. This chapter will discuss, independently, each of these internal stimuli.

### 8.1. The Primary Architecture of the Self-Assembling Dendron or Dendrimer and the Mode of Its Attachment to the Polymer Backbone

Figure 14 [187] outlines the diversity of possibilities for the attachment of a self-assembling dendron or dendrimer to a covalent or supramolecular backbone. They will only briefly be enumerated here. A self-assembling dendron can be attached covalently directly from its apex to the covalent backbone (a). Alternatively, it can be connected from apex to the backbone via a flexible spacer (b) or via a supramolecular interaction (c). For case (a), the covalent backbone can be replaced with a supramolecular backbone (d). The covalent attachments from (a) and (b) can be replaced with a covalent attachment directly from the periphery of the dendron without (e) or with a flexible spacer (f). Finally, twin self-assembling dendrons can be attached from their periphery to the backbone (g). Janus amphiphilic self-assembling dendrons can be attached via one (h) or the other side of their amphiphilic structure (i). Depending on the mode of attachment and of their structure the resulting self-organizable dendronized polymers can form spherical, helical cylindrical (Figure 2, right-side and Figure 15), bundles of cylinders surrounding the backbone that can result in periodic arrays [121,122] or superlattices [159]. In this Perspective, we will limit our discussion to cases (a), (b), (c), and (d) from Figure 14.

### 8.2. Secondary Structure of the Self-Assembling Dendron. Tapered and Conical

Tapered self-assembling dendrons attached to the polymer backbone directly or via a flexible backbone as shown in Figure 14a,b induce a helical conformation in the backbone that fits the helical supramolecular column that is jacketing it (Figure 2, right side and Figure 15) [22,23,24,25,26,27,28,29,30,31,32,33,34,35,36,37,38,39,40,41,42,43,44,45,46,47,48,49,50,51,52,53,54,55,56,57,58,59,121,180,181]. By analogy, a conical dendron will self-assemble a supramolecular sphere with the backbone adopting most probably a random coil or also a helical conformation [121]. A maximum degree of polymerization can be tolerated by the spherical supramolecular polymer [121,122]. Induction of a helical backbone conformation by a helical supramolecular column jacketing was demonstrated, and it will be discussed in more detail later in this Perspective. However, the definitive conformation of the polymer backbone in a supramolecular sphere was not yet definitively demonstrated.

### 8.3. Degree of Polymerization

Any degree of polymerization is tolerated by self-organizable dendronized polymers containing tapered side-groups that end up being coated in supramolecular helical dendrimers. However, the degree of polymerization of the conical attached dendron self-assembling a spherical dendrimer is limited to the degree of polymerization that can be incorporated into the sphere [121]. Depending on the polymerization temperature and the mechanism of polymerization, a self-interruption of the polymerization process can occur exactly when the last conical dendron completes the assembly of the supramolecular sphere leading to the construction of monodisperse polymers by a self-interruption process [123]. However, at high temperatures, the conical dendron is quasiequivalent and therefore, it undergoes a shape change from conical to tapered, transforming the structure of the polymer from a sphere to a column [121]. Figure 16 illustrates the process mentioned above for the case of the living ring-opening cationic polymerization of a cyclic imino ether or oxazoline monomer, and the structures of the resulting polymers resulted in different degrees of polymerization as demonstrated by X-ray diffraction experiments [144]. The experiment illustrated in Figure 15 was performed with other monomers attached to conical dendrons including acrylates, methacrylate, styrene, etc. [121].

### 8.4. Stereochemistry of the Polymer Backbone

When a less flexible backbone adopts a helical conformation determined by its stereochemistry, the stereochemistry of this backbone is very important. We will discuss in more detail the case of dendronized poly(phenylacetylene) and of other poly(arylacetylene)s. This concept will be discussed with the help of Figure 17. 

Cis-cisoidal and cis-transoidal poly(phenylacetylene) (PPA) exhibit compact and less compact helical conformations in solid state [96,97,98,99,100,101,188,189,190,191,192,193,194,195,196,197,198,199,200]. In solution, a helix–coil conformational transition occurs. This transition is accompanied at high temperatures by an intramolecular electrocyclization followed by irreversible chain cleavage (Figure 17a,b). When the backbone of the cis-transoidal or cis-cisoidal PPA is coated with a self-assembling dendron that fits the conformation of helical conformation of PPA—the helix-coil transition accompanied by electrocyclization is eliminated and a new helix–helix transition is generated. This reversible transition can be monitored by circular dichroism in solution and film and by X-ray diffraction experiments performed on oriented fibers. At low temperature, a compact cis-cisoidal helix is generated that undergoes a new helix–helix transition to a more extended cis-transoidal helix at high temperatures. A decrease in the diameter of the oriented fiber is accompanied by an extension of the fiber producing a molecular machine that lifts coins. The second part of this concept is illustrated in Figure 17a,c–e.

## 9. External Stimuli

By analogy with internal stimuli that refers to any structural molecular parameter that is internal to the architecture of the supramolecular dendrimer, external stimuli refer to all stimuli that are external to the structure of the supramolecular dendrimer. Environment, temperature, light, electric, magnetic fields, and many others are external stimuli that will be discussed in the next subchapters.

### 9.1. Environment

Environment can refer to stimuli that are present in solid state or bulk state, solution and even stimuli that are present in the gas phase including in the air. Selected examples of all these stimuli will be briefly discussed.

#### 9.1.1. Solid-State or Bulk-State Stimuli

##### Electron-Donor and Electron-Acceptor Components as External Stimuli

We will start the discussion of the bulk state or solid-state stimuli with donor and acceptor components that can interact in solid state with a supramolecular dendrimer and affect its behavior. Figure 18 [201,202,203] summarizes this concept that was elaborated in our laboratory. Self-assembling dendrons functionalized with an electron-donor or electron-acceptor at their apex may self-assemble or not into supramolecular helical columns organizing their donor or acceptor in the center of the helical column. In their native crystal or amorphous state these acceptors and donors may exhibit poor charge electron mobilities. However, after self-assembly in the center of a helical column, they exhibit excellent charge carrier mobilities that can be many orders of magnitude higher than in the native state. In the presence of an external donor or acceptor, simple contact between the donor dendron or acceptor dendron with its complementary component will form a donor–acceptor complex in a solid state that will incorporate the complex in the center of the helical column. Alternatively, donor dendrons will co-assemble with acceptor dendrons in a solid state or by simple grounding in a mortar and co-assemble a donor–acceptor dendron–dendron complex in a helical supramolecular column. The most powerful event will be when an amorphous polymer containing donor or acceptor side groups will be mixed in the solid state with the corresponding complementary acceptor dendron or donor dendron. A co-assembly process will occur placing the amorphous donor of the acceptor polymer in a helical conformation in the center of the supramolecular column via donor–acceptor interactions. Charge carrier mobilities for the amorphous polymers are very low. However, the incorporation of the amorphous polymer into a helical columnar arrangement also increases the charge carrier mobility by several orders of magnitude. Backfolded structural defects, as shown in the top columnar hexagonal arrangement from the top right side of Figure 18, are self-repairing upon heating and subsequent cooling. In this case, donor–acceptor external stimuli are mediated and self-repaired by temperature as second external stimuli. This concept also belongs to parts (c,d) from Figure 14.

##### Metal Salts and Their Interactions with Crown Ethers or PEG Attached to the Apex of a Self-Assembling Dendron

Related external stimuli are provided by solid metal salts. Metal salts are recognized by crown ethers (CE) or by podants that are poly(ethylene glycol)s of various degrees of polymerization (PEG). Self-assembling dendrons containing CE or PEG at their apex spontaneously co-assemble with metal salts and incorporate the salt in the center of a helical column creating a supramolecular polymer. Numerous examples were designed by our laboratory by employing this strategy [41,42,56,113,154,204,205,206]. The salt dissolved in the center of the column becomes an ionic conductor whose mobility is lower in the crystal state and increases dramatically in the liquid crystal state. Figure 19 provides an example illustrating this concept mediated by PEG at the apex of the self-assembling dendron. The thermal stability of the supramolecular column increases by increasing the amount of salt in its supramolecular polymer [41,42,56,112,113,204,205,206]. Also, a combination of dendrons that does not self-assemble in the presence of salt and its co-assembly by complexation with salt must be mentioned. A non-self-assembling dendron containing a crown ether at its apex is melted on a glass plate located on an optical polarized microscope. A few crystals of sodium or potassium triflate are dropped on it to see the spontaneous transition from an isomorphic black liquid on the optical microscope to the formation of a focal conic texture, demonstrating the spontaneous self-assembly of the supramolecular column.

#### 9.1.2. Solvents

Solvents as stimuli will be classified as organic and aqueous solvents.

##### Organic Solvents

Organic solvents can be classified in the simplest way as solvophobic solvents for a certain dendron or supramolecular dendrimers or as good or ideal solvents for a certain dendron. Solvophobic solvents select a certain dendron secondary structure and subsequently, mediate self-assembly in solution. We will discuss briefly only several examples. Ideal or good solvents for a dendron do not select a certain secondary structure that mediates self-assembly, whereas a solvophobic solvent has the capability to select it.

##### Solvophobic Solvents

The top part of Figure 13 demonstrates what a solvophobic solvent’s role is. In an ideal solution or good solvent like chloroform, for example, the dendritic dipeptide exhibits a dynamic equilibrium mixture of its anti and gauche conformers that form an ideal solution. A solvophobic solvent like cyclohexane or hexane shifts the anti-gauche equilibrium to an all anti- or trans conformation for the benzyl ether groups of the dendritic dipeptide and the dendritic dipeptide self-organized the helical columns illustrated under their structures. Cyclohexane and hexane mimic the structure of the hydrophobic part of a cell membrane and, therefore, this self-assembly process occurs also in the bilayer of the cell membrane or of a cell membrane mimic allowing transport of water from the exterior to the interior of the cell of the hydrophobic channel of the helical porous assembly to be measured. Solvophobic solvents are commonly used to mediate self-assembly in solution.

##### Aqueous Solvents

Aqueous solvents can be water of buffers of different pH. Amphiphilic molecules, including phospholipids, mixtures of phospholipids with cholesterol, and PEG-conjugated phospholipids, self-assemble in water or buffer into vesicles named liposomes [207] (Figure 20) and stealth liposomes [208,209]. Amphiphilic block-copolymers self-assemble into vesicles named polymersomes [210]. Amphiphilic Janus dendrimers self-assemble into unilamellar [211] or onion-like dendrimersomes [212]. Glycopolymers form the simplest mimic of the glycan from the surface of cell membranes [213], glycodendrimers for more precise examples of glycan mimics [214], whereas glycoliposomes form also a complex mimic of the glycan [215]. The simplest approach to the glycan of the cell membranes was obtained by the self-assembly of amphiphilic Janus glycodendrimers [216]. Amphiphilic Janus glycodendrimers self-assemble as their counterpart Janus dendrimers can for both unilamellar and onion glycodendrimersomes [217]. The unique feature of Janus dendrimers and Janus glycodendrimers is that they can self-assemble in water or buffer into unimolecular and onion-vesicles by simple injection in water forming monodisperse unilamellar or onion-like dendrimersomes and glycodendrimersomes with predictable dimensions. All other amphiphilic building blocks shown in Figure 20 require very complex methods for self-assembly followed by fractionation in order to produce narrow dispersity vesicles. For numerous applications, the stability of vesicles in buffer at different pH values is requested.

##### Sugar Binding Proteins by Sequence-Defined Glycodendrimersomes in Aqueous Solvents

Sugar-binding proteins, known as lectins and galectins, interact with the glycan of the cell membranes and with the glycan mimics of the cell membranes. Numerous experiments on this topic were performed in our laboratory with glycodendrimersomes assembled from Janus glycodendrimersomes [217,218,219,220,221,222,223,224,225,226,227,228,229,230,231,232,233,234,235,236,237,238]. Glycodendrimersomes bind to sugar-binding proteins via their multivalency. We would like to mention here a single experiment performed towards the elucidation of the multivalency concept. This experiment was performed both with unilamellar and multilamellar onion-like glycodendrimersioimes and involves elucidation of the role of sequence-defined glycan mimic on the binding efficiency. Figure 21 illustrates these experiments. Sequence-defined Janus glycodendrimers were constructed as shown in Figure 21a,b. Their self-assembly into sequence-defined glycodendrimersomes provided from binding ability to a library of lectins is shown in Figure 21c. The lower the concentration of sugar in their glycan mimics the higher the activity towards sugar-binding proteins. These results contradict most of the literature results that predicted the highest activity to occur at the highest concentration of sugar. The highest activity at a low concentration of sugar was demonstrated to be due to a different rate of contact of interaction that was supported by a different activity of the glycan due to its different architecture or morphology [217,218,219,220,221,222,223,224,225,226,227,228,229,230,231,232,233,234,235,236,237,238]. 

##### Decoration of Dendrimersomes with Proteins

A different mechanism to attach proteins directly on the surface of dendrimersomes is by dendrimersomes decorated with Janus dendrimers functionalized with NTA. Janus-NTA developed in our laboratory [232,233] are very active in binding proteins via their histidine groups. The protein coat generated via this mechanism subsequently binds DNA to create a second coat of DNA on the surface of the dendrimersome. This process is illustrated in Figure 22 [232].

##### Hybrid Membranes by Co-Assembly of Dendrimersome and Glycodendrimersomes with Bacteria Membranes

Giant dendrimersomes and giant glycodendrimersomew were demonstrated to co-assemble with bacterial membrane vesicles creating hybrid vesicles transplanting lipoproteins, channel proteins, and other components of the bacterial membrane into the bilayer of the hybrid cells. This is a process that is difficult to accomplish via any other technology since transmembrane proteins are not soluble in water [223]. Figure 23 outlines this concept that is of interest for numerous applications.

##### Hybrid Membranes by Co-Assembly of Dendrimersomes with Human Cell Membranes

Figure 24 illustrates the mechanism of the co-assembly of dendrimersomes with vesicles assembled from human membrane vesicles. As shown in Figure 23 for the case of bacterial cell membranes, successful experiments were shown to co-assemble dendrimersomes with human cell membranes [219].

##### Encapsulation of Bacteria into Dendrimersomes

Figure 25 outlines the encapsulation of a living bacterial into a dendrimersome [233]. A proper combination of interaction between the dendrimersome and bacteria facilitates the encapsulation of a bacterial into a dendrimersome via endocytosis. This first demonstration of the encapsulation of a living bacteria into a synthetic membrane mimic is important for numerous applications that will not be discussed here. The encapsulated bacteria remains alive for a long period of time, fighting to exit the synthetic dendrimersome without success for as long as it has food to survive. We encourage the reader of this Perspective to consult the original publication that contains numerous movies demonstrating this concept.

##### Encapsulation of mRNA into Dendrimersome Nanoparticles and their Endocytosis into Living Cells, Release of mRNA into Living Cells, and Synthesis of Proteins in Collaboration with the Ribosome

This concept provides a new technology for one-component mRNA vaccines. The classic mRNA vaccines are accomplished via a four-component technology that involves a combination of phospholipids, cholesterol, phospholipids conjugated to PEG, and an ionizable amine that becomes protonated at an acidic pH and deprotonated at the physiological pH. Figure 26 outlines the concept of four-component and one-component delivery of mRNA systems [234,235,236,237,238,239,240]. A complex microfluidic technology is used at low pH with the four-component system to generate nanoparticles incorporating the mRNA in their interior via interaction of the phosphate groups of mRNA with the protonated ionizable amines of the four-component system. Fractionation, followed by dialysis to deprotonate the exterior of the nanoparticles is needed before the injection process. The two-component system consists of sequence-defined Janus glycodendrimers in which the carbohydrate was replaced with an ionizable amine. Injection of the ionizable Janus dendrimers together with mRNA at low pH self-assemble nanoparticles of predictable size with narrow dispersity. No microfluidic technology, fractionation, or dialysis is required, and the nanoparticles can be directly injected into mice. Upon endocyctosis to the cell, the mRNA is released and in combination with the ribosome will synthesize the required protein. Figure 26 outlines this concept that we will not describe in more detail.

#### 9.1.3. Temperature

Several examples were already discussed in Figure 13, Figure 15, Figure 16, Figure 17 and Figure 18, and we will not repeat them here. Aside from change ion conductivity, self-repairing processes, molecular machine, deracemization, temperature provides a new methodology to assemble unprecedently new combinations of helical supramolecular dendrimers via the supramolecular orientational memory effect (SOM) [86,175,176,177,178,179,240,241,242,243,244] is summarized in Figure 27. 

SOM occurs at the transition from a Frank–Kasper supramolecular spherical dendrimer generated only from crown conformations of dendrimers to columnar assemblies and back and is nucleated by the closest distance spheres of the Frank–Kasper phase. Recently, this concept was demonstrated to provide the only methodology for the discrimination between different mechanisms of self-assembly when X-ray analysis is inadequate [86].

#### 9.1.4. Light

Photocleavable supramolecular dendrimers were investigated and demonstrated a mechanism for their dis-assembly and reassembly of different size spherical dendrimers. This is an important concept with numerous applications in drug and other delivery processes [245,246]. Such a mechanism is briefly outlined in Figure 28. 

#### 9.1.5. Electric and Magnetic Fields

Figure 29 outlines the structure of one of the first self-organizable dendrimer reported from our laboratory: the “willow” dendrimer [247,248,249,250]. “Willow” dendrimers and trees do not adopt a spherical conformation even at high generations. 

They collapse into a bilayer structure when placed on the ground [247,248,249,250]. The concept shown in Figure 29 is self-explanatory. The gauche conformation of the repeat unit from the top of the Figure becomes anti and the corresponding dendrimer exhibits classic smectic and nematic thermotropic mesophases. What is not trivial from this Figure and is not available in the original publications is that the switching time of this dendrimer under a magnetic field is higher than that of most classic thermotropic liquid crystals. The reason for this process is not yet known and, therefore, it was not yet published, but we think it is important to be disclosed here.

## 10. Conclusions

The inspiration for this Perspective was initiated by the guest editor of this special issue, Professor Ivan Gitsov. Without his invitation we would have never thought about writing a Perspective on stimuli-responsive complex self-organized dendrimers, and in addition, consider the topic in the way we reported stimuli-responsive dendrimer systems here. In order to do so, we had to neglect numerous contributions to this field including many from our own laboratory [251]. We also had to neglect very important contributions to this field from the laboratory of the guest editor [252,253,254,255,256]. As we have mentioned in the Abstract, Introduction, and in the rest of this Perspective, we preferred to discuss only topics of research for which we consider that we were qualified to provide a different view on stimuli-responsive principles. We also tried to avoid being influenced by other excellent review articles on the topic of stimuli-responsive systems [257,258,259,260,261]. We hope that this Perspective will bring a complimentary view to all published work on this topic [257,258,259,260,261] and to other reviews and publications to be reported in this special issue. We also expect that this Perspective will impact other fields of soft and living self-organized matter as it was already demonstrated by our laboratory for the case of helical self-organizations [262,263]. As elegantly pointed out by Lehn, there are unlimited fields of self-organization [264] where these concepts can, and we hope will, provide an impact.

## Figures and Tables

**Figure 2 polymers-15-01832-f002:**
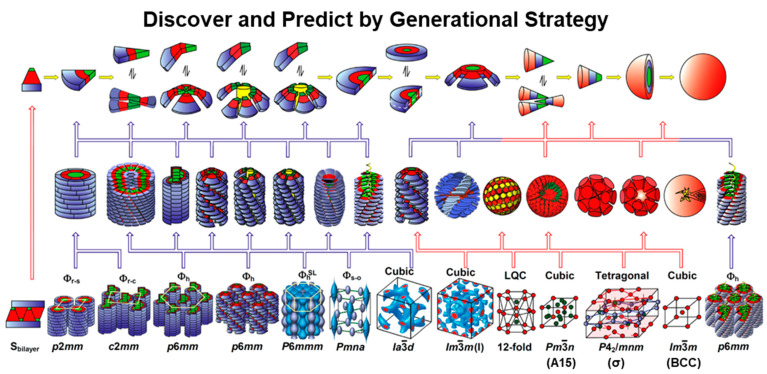
Structural and retrostructural analysis of supramolecular assemblies formed via the self-assembly and self-organization of dendrons help to discover and predict by generational strategy [60]. The Figure was adapted and modified from [60]. Reproduced with permission from [60]. Copyright © 2009, American Chemical Society.

**Figure 3 polymers-15-01832-f003:**
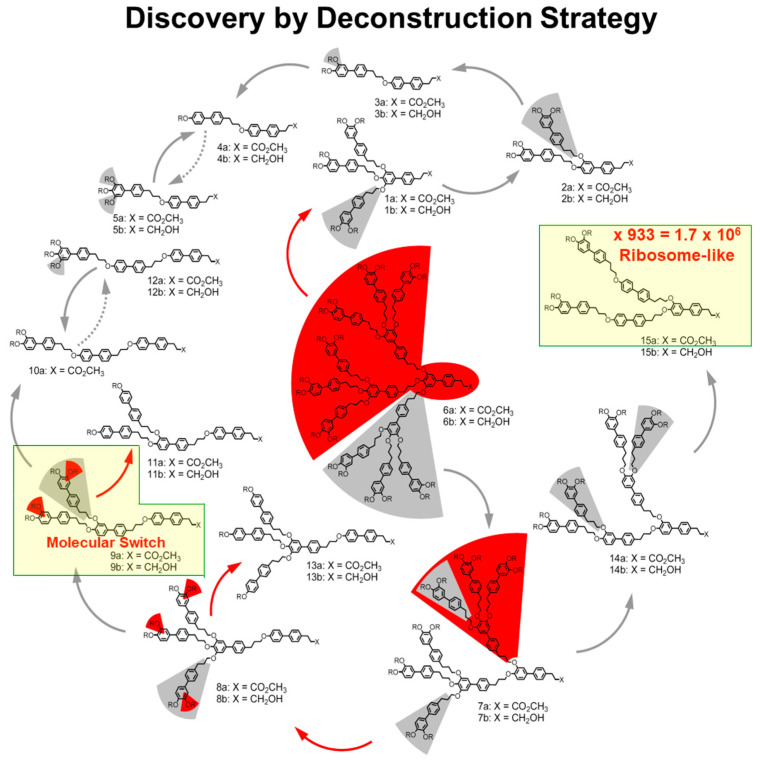
The deconstruction strategy of third-generation dendron (3,4BpPr-(3,4,5BpPr)_2_)12G3-X (shown in the middle) led to the discovery of new self-assembling minidendrons marked in yellow, where X = –CO_2_CH_3_ for 6a and –CH_2_OH for 6b. In each step of the deconstruction (solid arrows), the fragment highlighted by the wedge of the corresponding color (red or gray) is removed, and the remaining unmarked dendron is synthesized and structurally analyzed. Yellow highlights form a molecular switch and a ribosome-size supramolecular dendrimer [135]. The Figure was adapted from [135] and modified. Copyright © 2010 WILEY-VCH Verlag GmbH & Co. KGaA, Weinheim.

**Figure 4 polymers-15-01832-f004:**
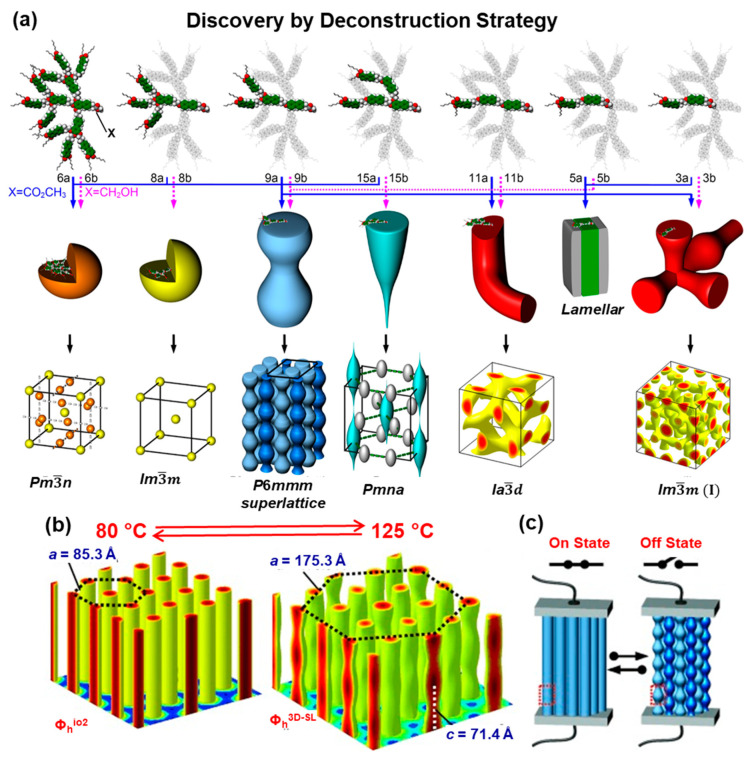
(**a**) A diversity of supramolecular assemblies generated from the primary structures designed by the deconstruction strategy. Novel structures from deconstructed dendritic esters (blue arrows) and alcohols (purple dotted arrows). (**b**) Reconstructed relative electron density distributions of the Φ_h_ and Φ_h_^3D-SL^ phases generated from 9b indicating the reversible transition with temperature. (**c**) The potential role as a supramolecular switch. [135]. The parts of the Figures were adapted from [135] and modified. Copyright © 2010 WILEY-VCH Verlag GmbH & Co. KGaA, Weinheim.

**Figure 5 polymers-15-01832-f005:**
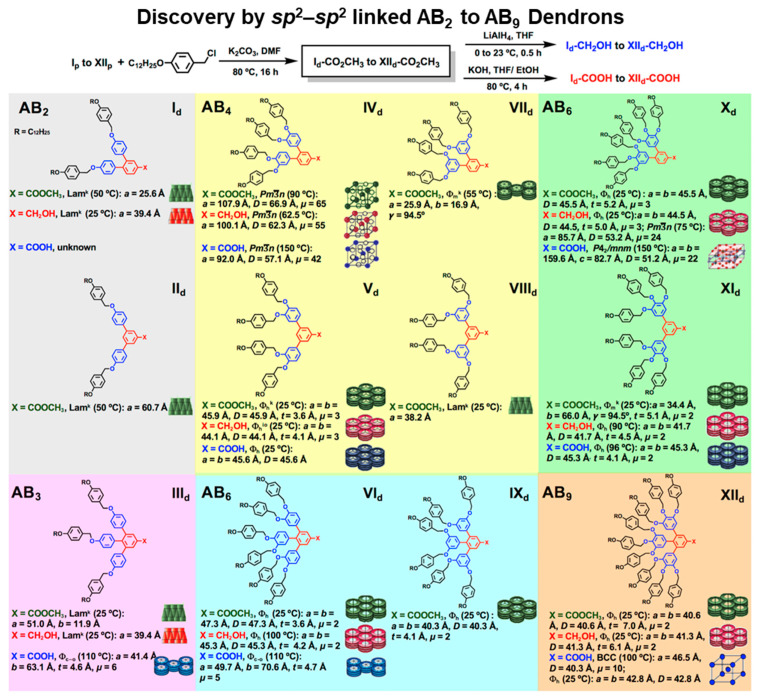
Structural and retrostructural analysis of supramolecular dendrimers self-assembled from a library of symmetric constitutional isomeric AB_2_ to AB_9_ self-assembling *sp*^2^-*sp*^2^ dendrons [86]. Reproduced with permission from [86]. Copyright © 2021, American Chemical Society.

**Figure 6 polymers-15-01832-f006:**
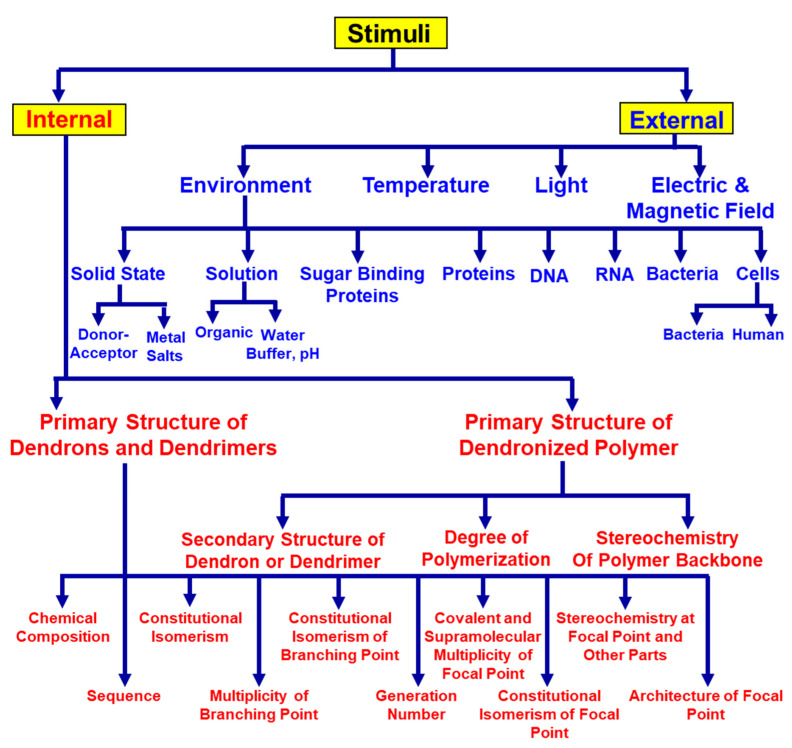
The classifications of internal- and external-stimuli in self-organized dendrimers generated from self-assembling and self-organizable dendrons, dendrimers, and dendronized polymers.

**Figure 7 polymers-15-01832-f007:**
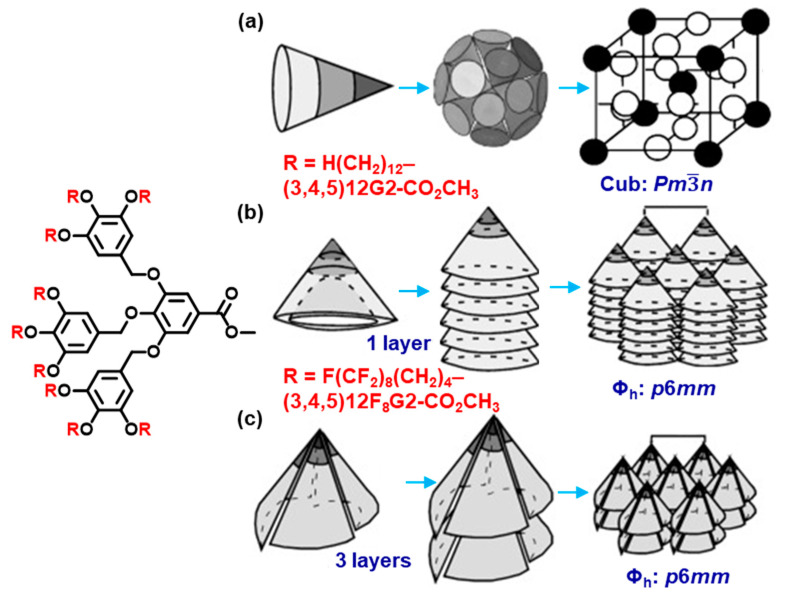
Chemical composition affecting the structure of self-organized supramolecular dendrimers. Structures, self-assembly, and self-organization of the second generation dendrons: (**a**) all-trans cone conformation of (3,4,5)^2^12G2-CO_2_Me; (**b**) all gauche-crown conformation of (3,4,5)^2^12F8G2-CO_2_Me; (**c**) all trans-taper conformation of (3,4,5)^2^12F8G2-CO_2_Me [136]. Parts of this Figure were adapted from [136] and modified. Copyright © 2003 WILEY-VCH Verlag GmbH & Co. KGaA, Weinheim.

**Figure 8 polymers-15-01832-f008:**
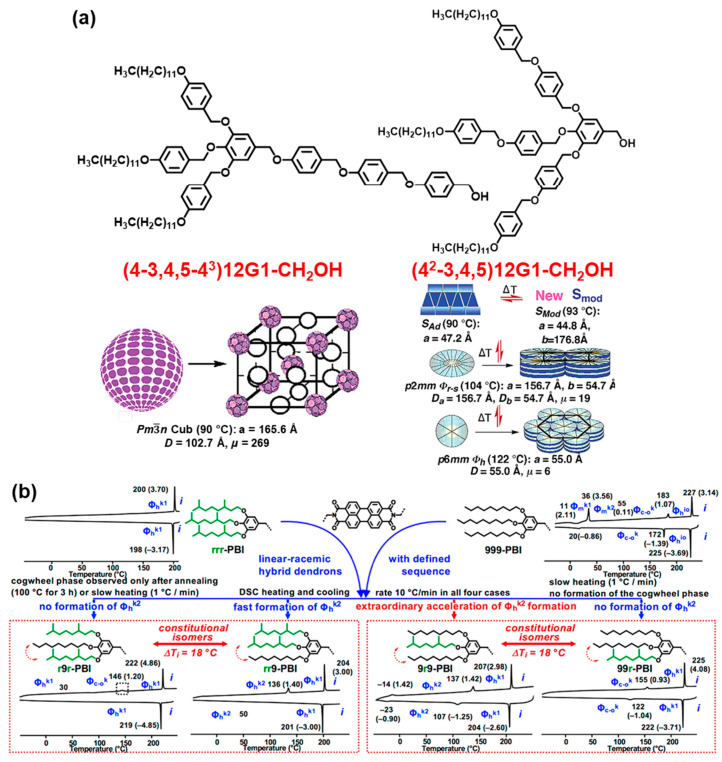
(**a**) Dendrons with identical chemical composition but different sequence forming different kinds of supramolecular structures. Supramolecular dendrimers self-assembled from (4-3,4,5-4^3^)12G1-CH_2_OH and (4^2^-3,4,5)12G1-CH_2_OH. (**b**) An example in which constitutional isomeric structures with identical composition but a different sequence of a combination of linear *n*-nonyl and branched chiral or racemic groups provide different supramolecular assemblies with different rates of heating and cooling. DSC traces of PBIs with sequence-defined hybrid r/*n*-nonyl dendrons were recorded upon second heating and first cooling at 10 °C/min. Phases determined by fiber XRD, transition temperatures (in °C), and associated enthalpy changes (in parentheses, in kcal/mol) are indicated. Phase notation: Φ_h_^k1^, columnar hexagonal crystal with offset dimers; Φ_h_^k2^, columnar hexagonal crystal with cogwheel assembly; Φ_c–o_^k^, columnar centered orthorhombic crystal; Φ_m_^k^, columnar monoclinic crystal, *i*, isotropic liquid [127,140]. Parts of the Figure were adapted, combined, and modified from [127,140]. Copyright © 2004, American Chemical Society. Copyright © 2020, American Chemical Society.

**Figure 9 polymers-15-01832-f009:**
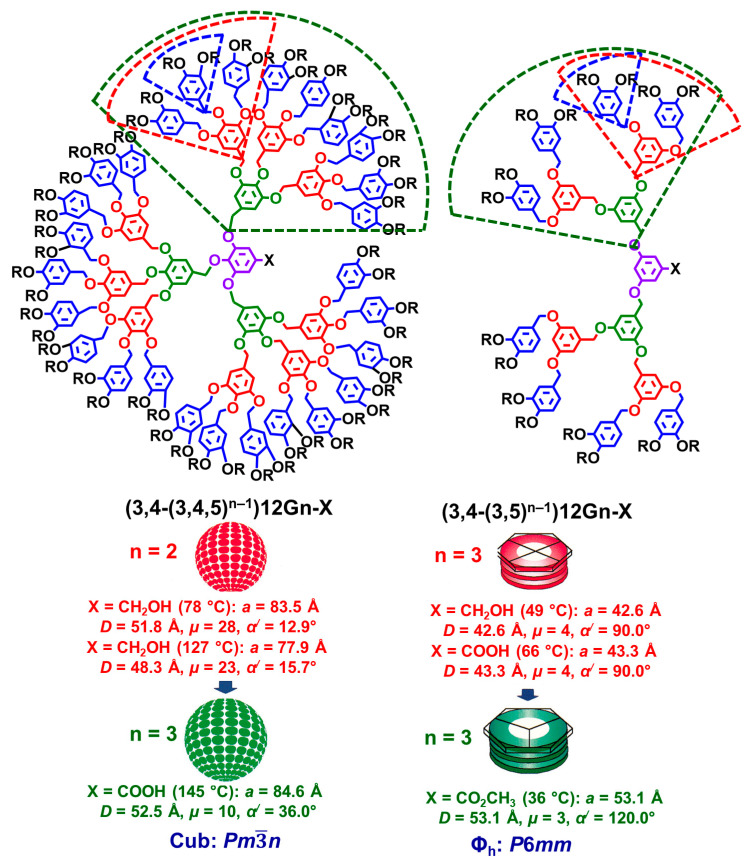
An example that demonstrates the role of the multiplicity of the branching point during the self-organization of a supramolecular assembly. Structural and retrostructural analysis of supramolecular dendrimers self-assembled from AB_3_ 3,4,5-trisubstituted monodendrons of generation 2 and 3 are shown on the left side and from a generation 3 AB_2_ 3,5-disubstituted dendron on the right side [126]. Parts of the Figure were adapted, combined, and modified from [126]. Copyright © 2004, American Chemical Society.

**Figure 10 polymers-15-01832-f010:**
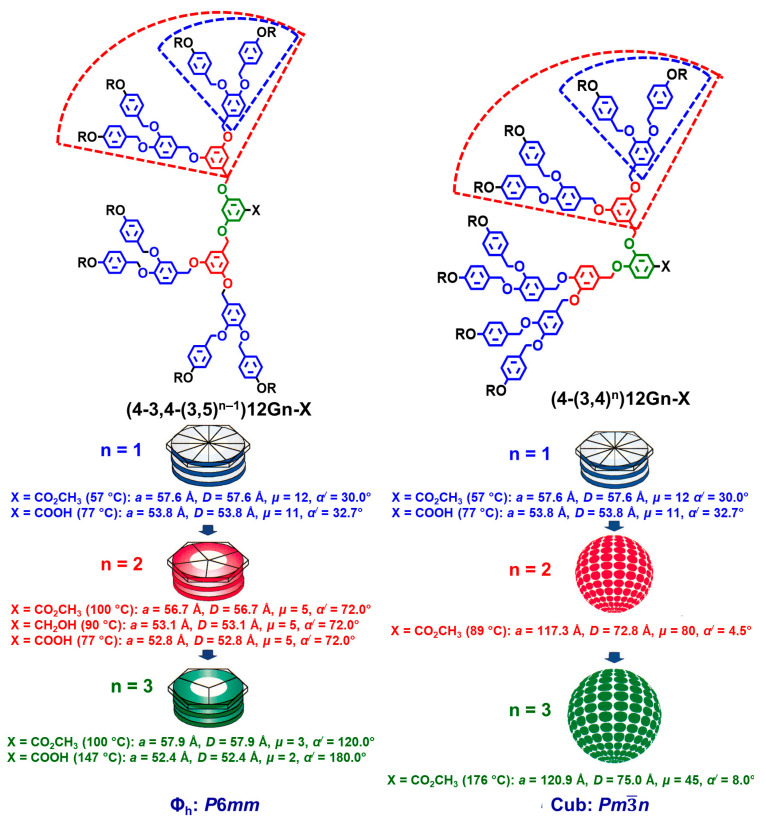
An example demonstrating the constitutional isomerism of the branching point affecting the supramolecular organizations. Structural and retrostructural analysis of supramolecular dendrimers self-assembled from AB_2_ 3,5-disubstituted dendron is shown on the left side and of its constitutional isomeric AB_2_ 3,4-disubstituted dendron is shown on the left side [126]. Parts of the Figure were adapted, combined, and modified from [126]. Copyright © 2004, American Chemical Society.

**Figure 11 polymers-15-01832-f011:**
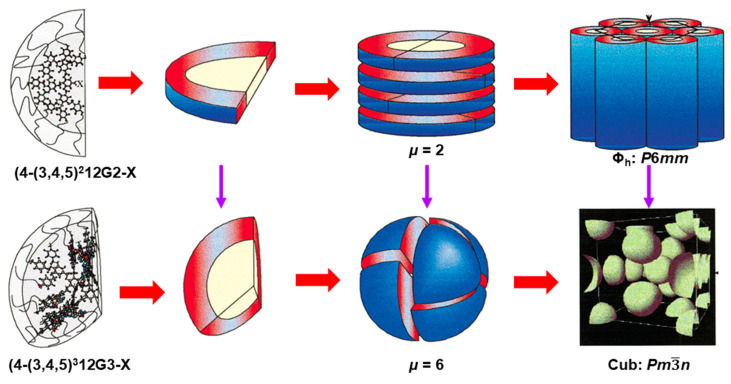
Hierarchical self-organization of the same self-assembling dendron at two different generations. (4-(3,4,5)^2^12G2-X is second generation (top), and (4-(3,4,5)^3^12G3-X is third generation (bottom) self-assembling dendron self-assembling columnar and, respectively, spherical assemblies generating columnar hexagonal and cubic *Pm*$\overset{-}{3}$*n* or A15 Frank–Kasper phases [124]. Parts of the Figure were adapted and modified from [124]. Copyright © 1998, American Chemical Society.

**Figure 12 polymers-15-01832-f012:**
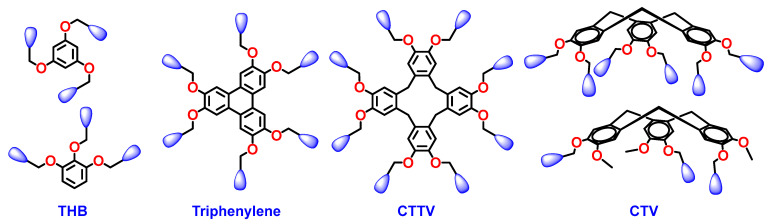
Examples of constitutional isomerism of the focal point. 1,3,5-trihyhdroxybenzene (THB), triphenylene, cyclotetraveratrylene (CTTV), cyclotriveratrylene (CTV).

**Figure 13 polymers-15-01832-f013:**
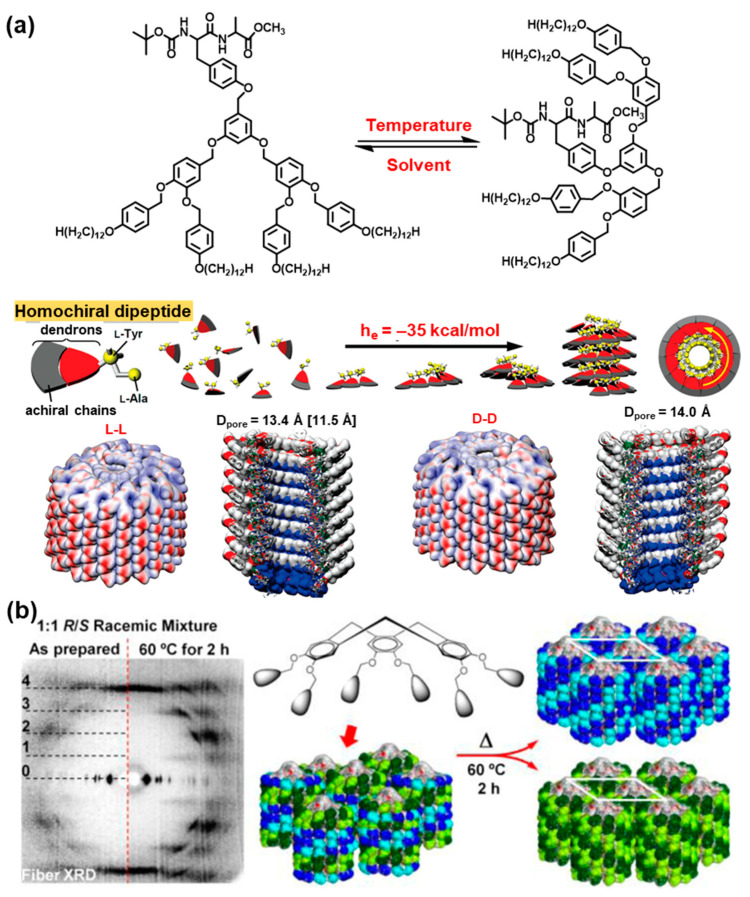
(**a**) Illustration of the concept of the helical dendritic dipeptide and of its self-assembly into hydrophobic helical pores. Structure, conformation and hydrogen-bonding of (4-3,4-3,5)12G2-CH_2_-Boc-L-Tyr-L-Ala-OMe and (4-3,4-3,5)12G2-CH_2_-Boc-L-Tyr-D-Ala-OMe during self-assembly. Self-assembly of homochiral dendritic dipeptides in porous supramolecular columns via a helical cooperative growth. Schematic and analysis of self-assembly via supramolecular polymerization in solution and the corresponding side-view and cross-section of the porous supramolecular columns determined from XRD analysis in the solid state. (**b**) A hat-like homochiral or racemic column is generated by a dendronized cyclotriveratrylene (CTV). The deracemization process between enantiomerically rich columns assembled from the hat-shaped dendronized CTV. Illustration of homochiral columns constructed by a chiral self-sorting supramolecular helical organization of hat-shaped molecules (S), (R)-(3,4Bn)dm8*G1-CTV; comparison of wide-angle XRD patterns before and after thermal annealing; demonstration of chiral self-sorting mechanism [161,167,171]. Parts of the Figure were adapted, combined and modified from [160,166,171]. Copyright © 2004, Macmillan Magazines Ltd. Copyright © 2011, American Chemical Society.

**Figure 14 polymers-15-01832-f014:**
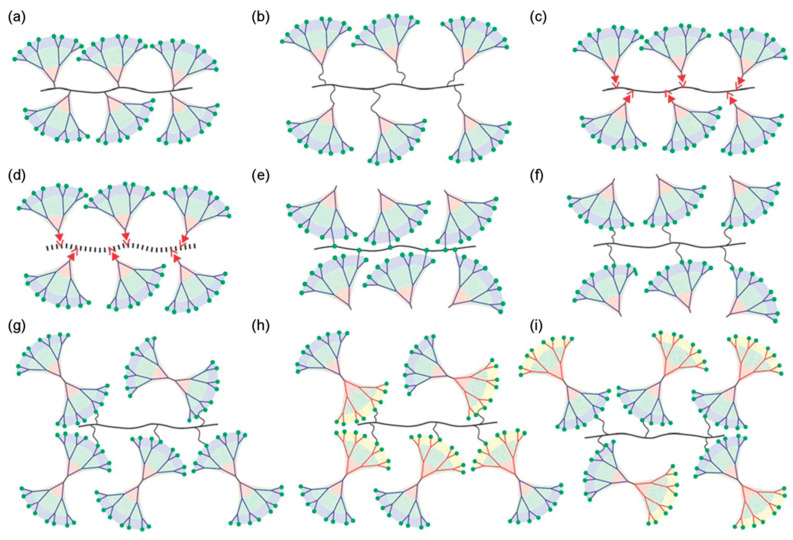
The diversity of possibilities for the attachment of a self-assembling dendron or dendrimer to a covalent or supramolecular backbone. Topologies generated from linear covalent and supramolecular polymers dendronized with self-assembling dendrons, twin dendritic molecules, and Janus dendrimers [185]. Dendron directly attached to the polymer backbone via apex (**a**), with a flexible spacer (**b**), attached via non-covalent interactions (**c**), supramolecular polymers dendronized (**d**), dendron directly attached to the polymer backbone via its periphery (**e**), via its periphery and a flexible spacer (**f**), covalent polymers dendronized with twin-dendrimers (**g**), and with Janus dendrimers (**h**,**i**). This Figure is reproduced with permission from [185]. Copyright © 2012, American Chemical Society.

**Figure 15 polymers-15-01832-f015:**
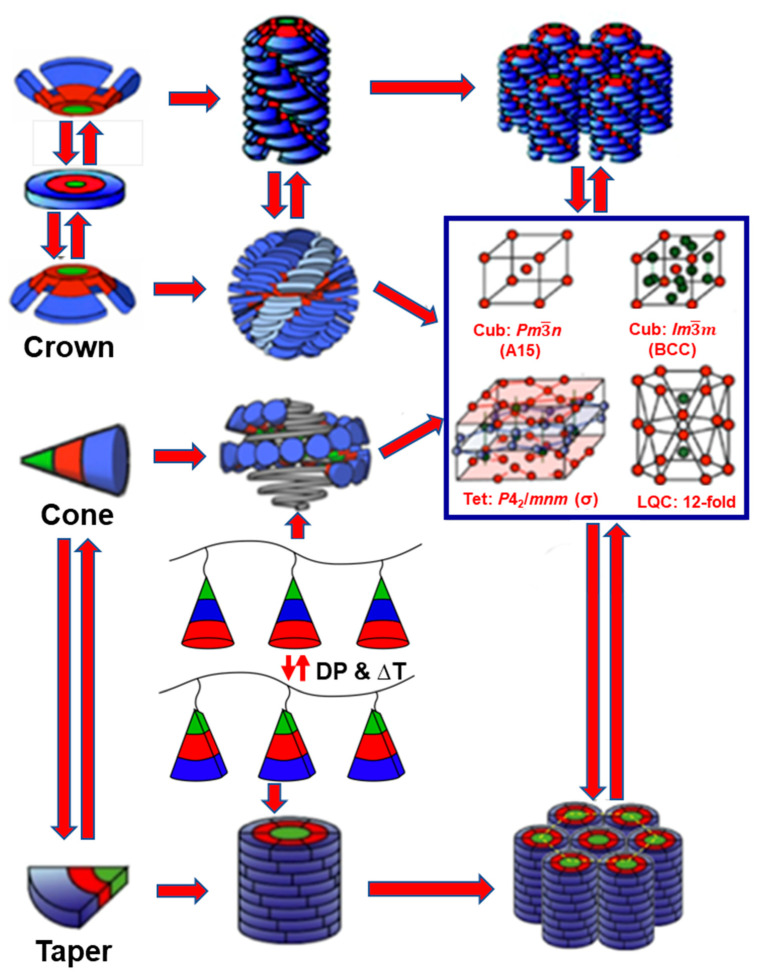
Self-assembly and self-organization of crown-like, cone-like and taper-like dendrons and other corresponding self-organizable dendronized polymers. Shape of secondary and tertiary structure mediated bt temperature and/or degree of polymerization of the dendronized polymer backbone.

**Figure 16 polymers-15-01832-f016:**
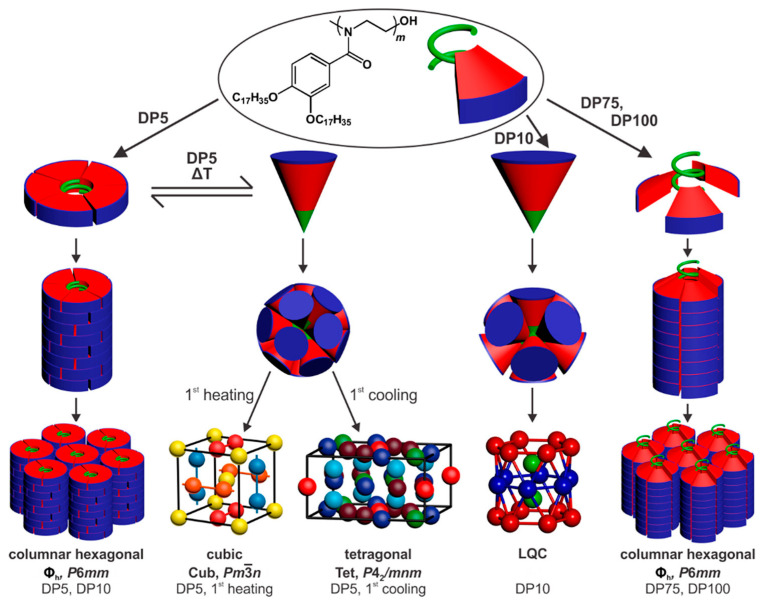
Summary of periodic and quasiperiodic arrays self-organized from assemblies of poly[(3,4)17G1-Oxz] at different degrees of polymerization (DP) and temperature [144]. This Figure is reproduced with permission from [144]. Copyright © 2018, American Chemical Society.

**Figure 17 polymers-15-01832-f017:**
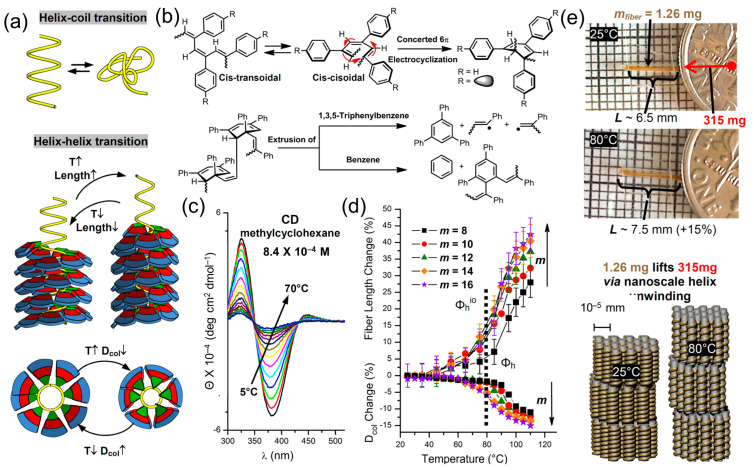
Molecular machine self-organized from dendronized helical polyphenylacetylenes. Illustration of the helix–coil transition and its transformation into a helix–helix transition that mediates expansion and contraction of the helical structure with temperature (**a**); expanded images collected by a digital camera at 25 °C and at 80 °C of the oriented fiber (**b**); variable-temperature CD spectrum (**c**); comparison of the fiber length change from optical microscopy and column diameter from the fiber XRD for the library of the polyphenylacetylenes with different peripheral alkyl chain length in the dendron (m) (**d**) [97,100]. The Figure is adapted from [97,100]. Copyright © 2005, American Chemical Society. Copyright © 2008, American Chemical Society.

**Figure 18 polymers-15-01832-f018:**
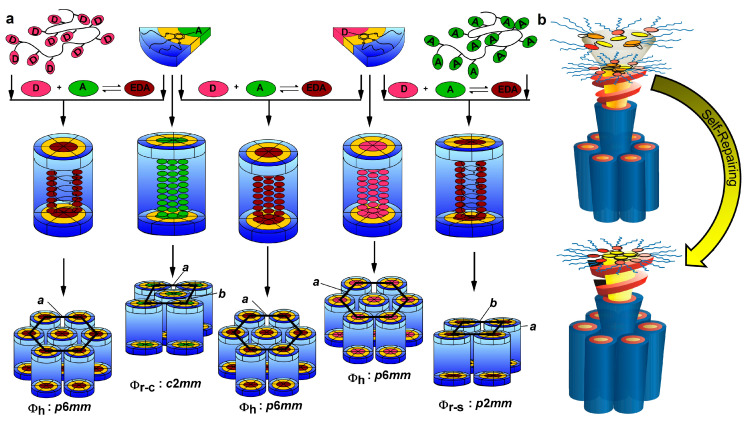
Schematic illustration of complex electronic supramolecular materials mediated by dendrons containing donor (D) and acceptor (A) groups, and their co-assembly with complementary amorphous polymers containing D and A side groups (**a**). The different systems form hexagonal columnar (Φ_h_), centred rectangular columnar (Φ_r-c_), and simple rectangular columnar (Φ_r-s_) arrays; a and b are lattice dimensions. The self-repairing process of back-folded (brown) electronically active supramolecular helical pyramidal columns self-assembled by semifluorinated minidendron attached to the acceptor groups (**b**) [200]. The Figure is adapted and modified from [201]. Copyright © 2002, Macmillan Magazines Ltd.

**Figure 19 polymers-15-01832-f019:**
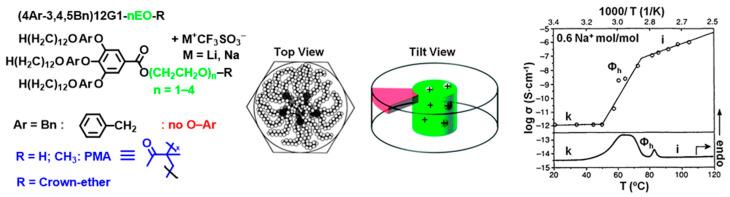
General molecular structure of dendritic monomers and dendronized polymers, and the top and tilt views of the supramolecular column [41,42,204]. The Figure was adapted and modified from reference [112]. Copyright *©* The Royal Society of Chemistry.

**Figure 20 polymers-15-01832-f020:**
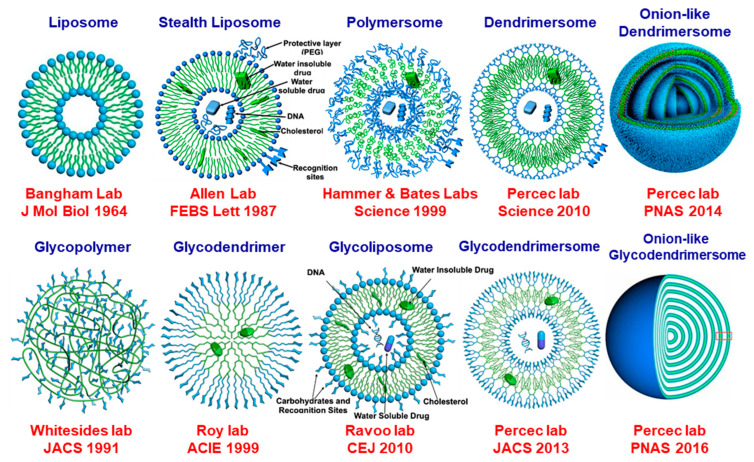
The development and structures of liposomes, stealth liposomes, polymersomes, dendrimersomes, onion-like dendrimersomes, glycopolymers, glycodendrimers, glycoliposomes, glycodendrimersomes, and onion-like glycodendrimersomes.

**Figure 21 polymers-15-01832-f021:**
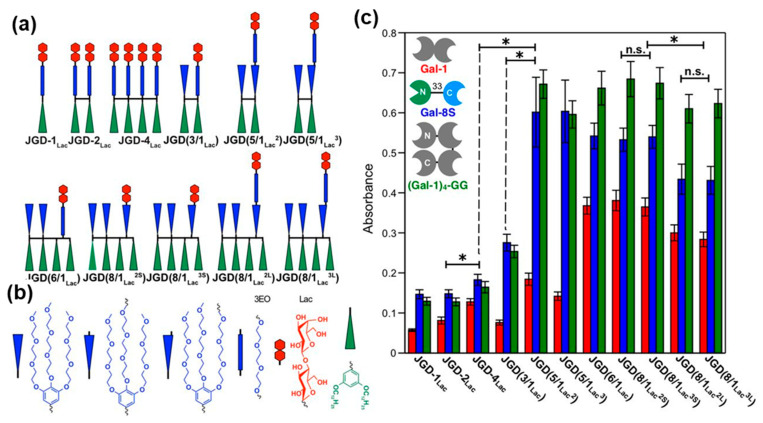
Encoding biological recognition in a bicomponent cell-membrane mimic. (a) Sequence-defined JGDs with different Lac densities, sequence, and linker length. (b) Schematic representation of JGD building blocks. (c) Summary of aggregation assay data using GDSs from self-assembly of sequence-defined JGDs (Lac = 0.1 mM, 900 μL) with Gal-1 (1 mg·mL^−1^, 100 μL), Gal-8S (1 mg·mL^−1^, 100 μL), and (Gal-1)4–GG (1 mg·mL^−1^, 100 μL). Color codes for galectins: Gal-1, red; Gal-8S, blue; (Gal-1)4–GG, green. N and C represent the N and C termini of proteins. For selected examples symbols used for significant difference (*p* values by Student’s t-test) are: “n.s.” for *p* > 0.05 (for statistically nonsignificant) and “*” for *p* < 0.05 (for statistically significant) [226]. Copyright (2019) National Academy of Sciences USA.

**Figure 22 polymers-15-01832-f022:**
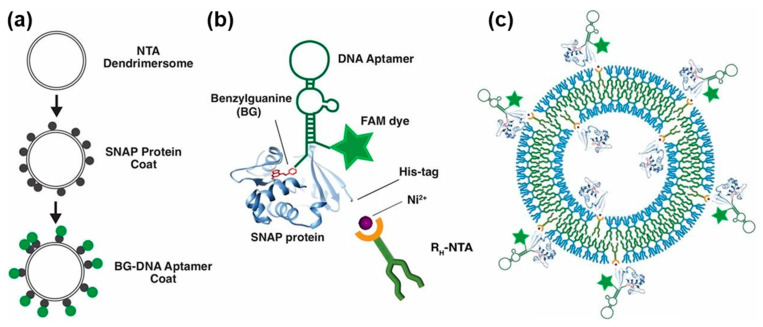
Modular tethering of DNA to SNAP-tagged dendrimersomes. (**a**) Schematic showing layering of a protein and DNA coat to dendrimersome vesicle. (**b**) His-SNAP proteins bind to RH-NTA to form the initial protein layer. SNAP binds to a BG conjugated to the DNA, allowing for the modular formation of a second layer, composed of nucleic acids. The DNA aptamer is labeled with FAM to enable imaging. (**c**) schematic of multilayer dendrimersome containing DNA and protein coat [230]. Copyright (2019) National Academy of Sciences USA.

**Figure 23 polymers-15-01832-f023:**
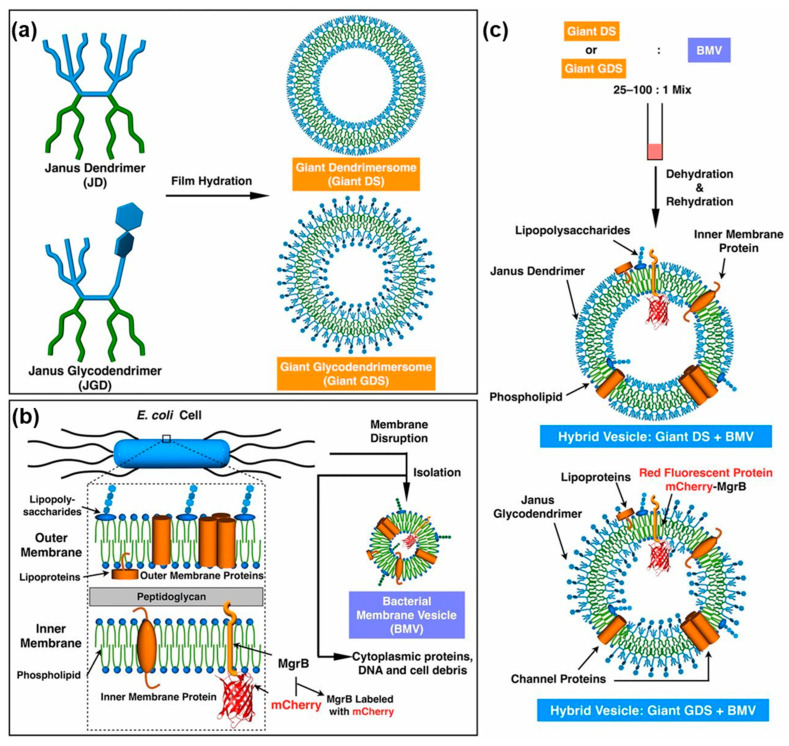
Illustration of (**a**) the preparation of giant DSs, (**b**) the preparation of BMV expressing YadA bacterial adhesin protein, and (**c**) coassembly of giant hybrid vesicles from giant DSs and *E. coli* BMV expressing YadA bacterial adhesin protein [222]. Copyright (2016) National Academy of Sciences USA.

**Figure 24 polymers-15-01832-f024:**
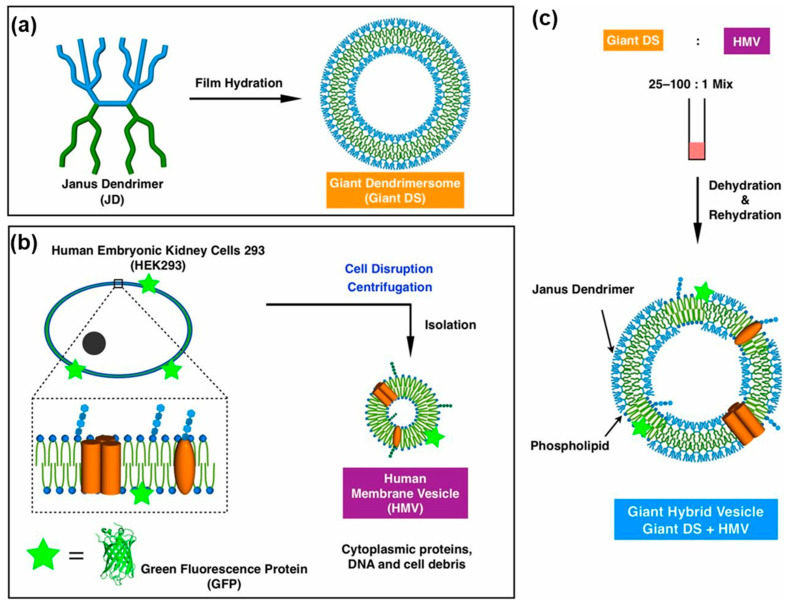
Schematic illustration of (**a**) the preparation of giant DSs, (**b**) the preparation of HMV from human kidney cells 293 (HEK293), and (**c**) coassembly of giant hybrid vesicles from giant DSs, and HMV from HEK293 labeled with GFP [218]. Copyright (2019) National Academy of Sciences USA.

**Figure 25 polymers-15-01832-f025:**
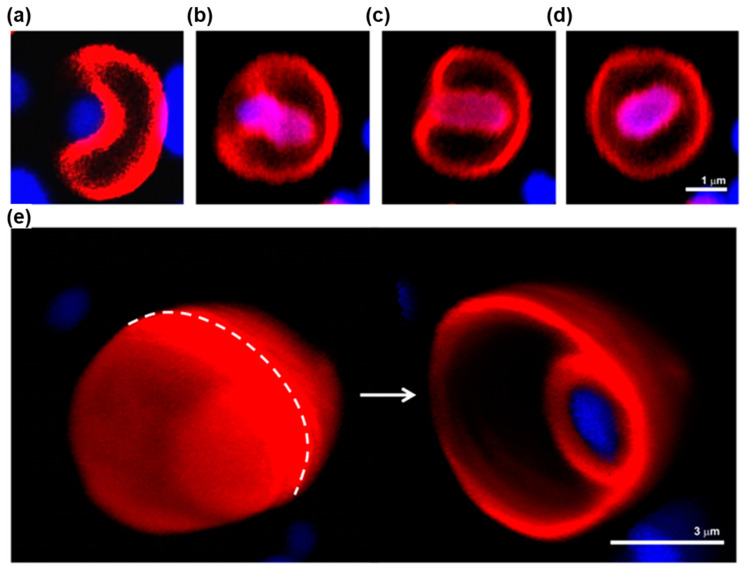
CLSM images show the process of engulfment of bacteria (blue) by DSs (red). (**a**) Adhesion of *E. coli* to the DS membrane. (**b**,**c**) Invagination of *E. coli* into the interior of the DS. (**d**) Formed endosome with living bacteria inside. (**e**) 3D reconstruction of 150 confocal scans for the whole (left) and 80 confocal scans for half (right) of the DS with engulfed *E. coli*. The white dashed line on a whole DS indicates the place of intersection for the presentation of half of the DS to show the interior of an endosome with engulfed bacteria [232]. The Figure was adapted from [233]. Copyright © 2019, American Chemical Society.

**Figure 26 polymers-15-01832-f026:**
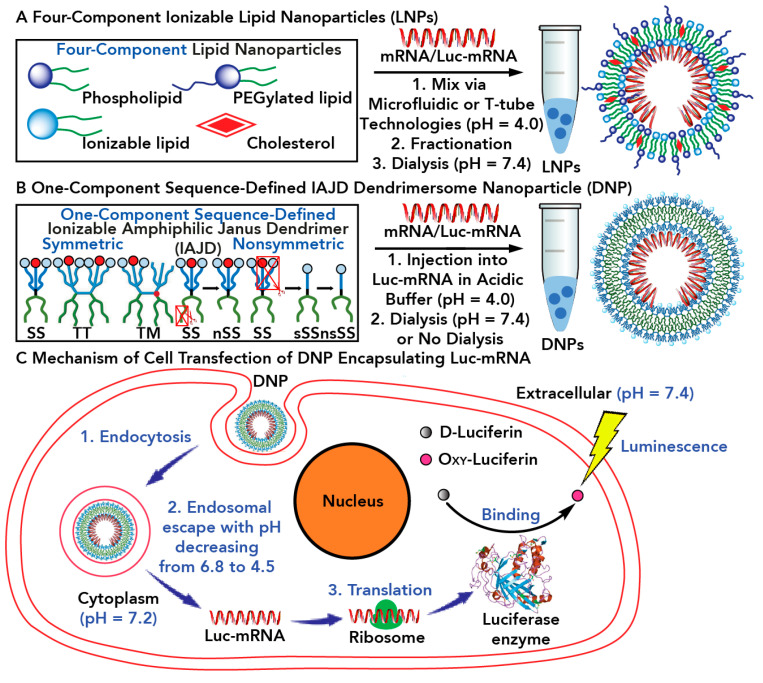
Schematic representation of four-component LNPs, one-component IAJDs based DNPs, and the cell transfection mechanism of LNPs and DNPs encapsulating Luc-mRNA (**A**) Four-component LNPs. (**B**) One-component IAJDs and their DNPs. (**C**) Cell transfection mechanism of LNPs and DNPs encapsulating Luc-mRNA.

**Figure 27 polymers-15-01832-f027:**
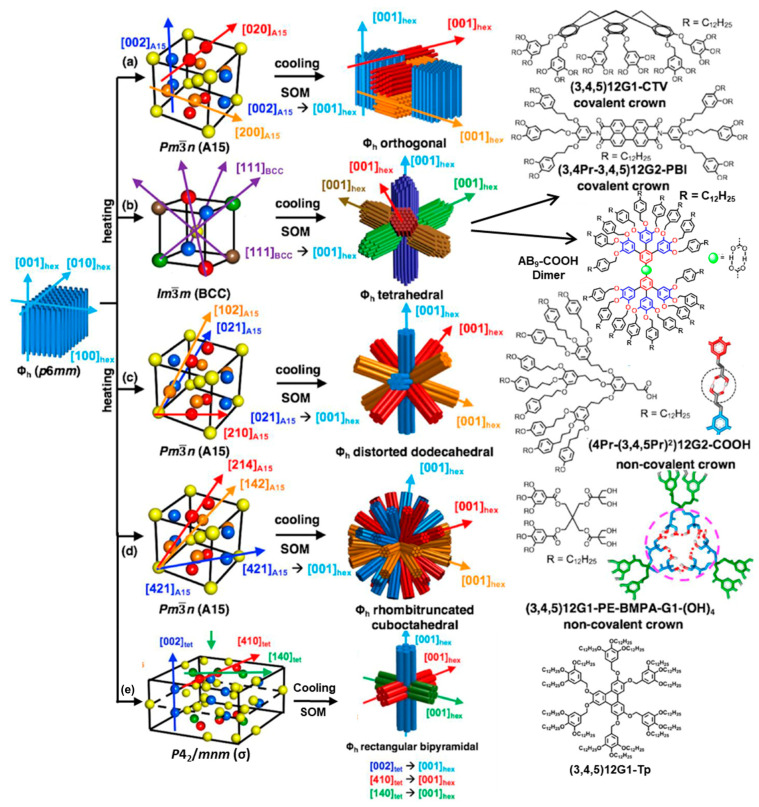
Summary of the supramolecular orientational memory (SOM) effect with selected examples of new bundles of columnar arrays and new molecular bundles of columnar hexagonal arrays generated by SOM [86,179]. SOM from Φ_h_ to (**a**) *Pm*$\overset{-}{3}$*n* (A15); (**b**) *Im*$\overset{-}{3}$*m* (BCC); (**c**) *Pm*$\overset{-}{3}$*n* (A15); (**d**) *Pm*$\overset{-}{3}$*n* (A15); and (**e**) *P*4_2_/*mnm* (σ). This Figure is reproduced with permission from [178]. © 2022 The Author(s). Published by Elsevier Ltd.

**Figure 28 polymers-15-01832-f028:**
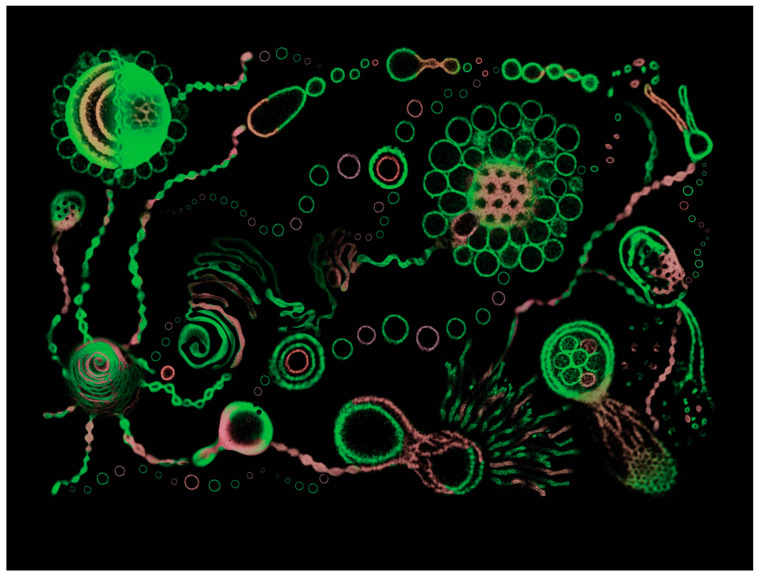
Mechanism of disassembly of large dendrimersomes by photocleavage and their reassembly into smaller dendrimersomes [245]. Copyright *©* The Royal Society of Chemistry.

**Figure 29 polymers-15-01832-f029:**
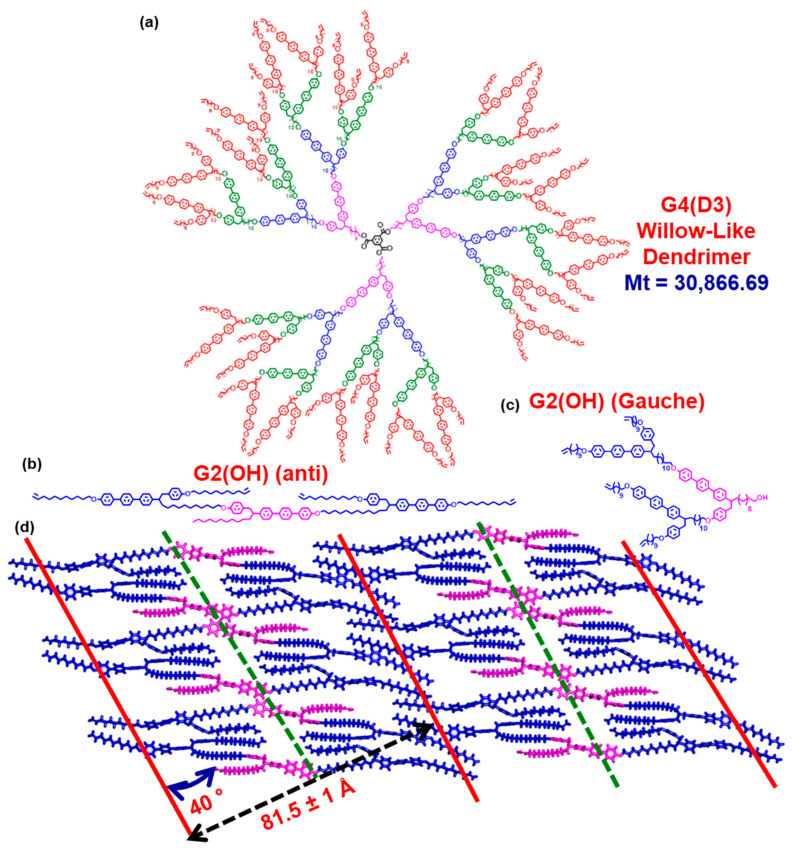
(**a**) Molecular structure of dendrimers **G4(D3)**; (**b**) Molecular structure of dendron **G2(OH)** (anti), and (**c**) Molecular structure of dendrimers **G2(OH)** (gauche). (**d**) Molecular model of the smectic phase of **G2(OH)**. The magnetic field is almost horizontal [249].

## Data Availability

Not applicable.

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
