# Peer review of "Stimuli-Responsive Principles of Supramolecular Organizations Emerging from Self-Assembling and Self-Organizable Dendrons, Dendrimers, and Dendronized Polymers"

_polymers, 2023, doi:10.3390/polym15081832_

Round 1
Reviewer 1 Report
Due to numerous potential applications in the field of catalysis, materials, biology and medicine, dendrimers constitute an important class of macromolecular compounds. Dendrimers are built step by step, by the repetition of a sequence of reactions, which allow multiplying the number of branches and the number of end groups. The number of peripheral units and the global shape are closely related to the nature of the core; molecule has mainly a three-dimensional tree-like structure, emanating from one centre – the core, with branches (dendrons). The size of such a molecule is determined by the generation number, i.e. by the length of the dendrons, which consist of an equal number of repeating units terminated by the end groups. Dendrimers have a tree-like three dimensional structure with exact numbers of repeating units and terminal groups, i.e. they are monodisperse compounds, thanks to their step-by-step synthesis.
This article describes and systematizes for the first time the principles of stimulating various types of supramolecular organizations arising from self-assembling and self-organizing dendrons, dendrimers and dendronized polymers. The history of the discovery of dendrimers and the conditions for their self-ordering into various supramolecular structures are briefly described. The patterns obtained are fulfilled not only in the chemistry of dendrimers, but are of interest for biology, materials science, and even for the social sciences.
The structures resulting from the self-ordering of dendrimers are not only of practical interest for obtaining liquid crystal materials, but can also be considered as objects of art. The authors rely entirely on their own pioneering development of strategies for obtaining various supramolecular structures. The paper presents a classification of various stimuli for the organization of dendrons for a wide range of dendrimers and dendrons.
This task is very relevant and its solution is new and original. In addition, the possibility of obtaining a huge number of dendrimer structures using various procedures has been studied. Overall, the results show that dendrimers are promising new materials for the production of liquid crystals and other structures.
The structure of supramolecular systems is confirmed by the data of various physical methods. The article presents a detailed discussion of the structure of the obtained complex systems based on dendrimers. The conclusions made in the work are consistent with the experimental data obtained and with the aim of the work. The article presents a fairly detailed list of references, which is informative and indicates that the authors are familiar with modern literature data.The language of the article is good.
I believe that the work can be recommended for publication in the Polymers Journal.
Author Response
Thank you very much for your high evaluation.
Reviewer 2 Report
This manuscript summarizes the stimuli-responsive principles of supramolecular organizations emerging from self-assembling and self-organizable polymers. The development and application of Dendrons, Dendrimers and Dendronized Polymers are introduced from shallow to deep. The content is comprehensive and well-organized. In addition, the informative content will be helpful for the research of polymers in these directions. Therefore, the manuscript potentially deserves publication in Polymers, but the following points have to be carefully reconsidered before publication.
1. Please explain the conceptual difference between Self-Assembling and Self-Organizable.
2. There is a problem with the formatting of lines 113 to 116, please revise it.
3. The format of “7.7 Covalent and Supramolecular Multiplicity of the Focal Point” is different from others, please revise it and check for the same type of problem.
Author Response
This manuscript summarizes the stimuli-responsive principles of supramolecular organizations emerging from self-assembling and self-organizable polymers. The development and application of Dendrons, Dendrimers and Dendronized Polymers are introduced from shallow to deep. The content is comprehensive and well-organized. In addition, the informative content will be helpful for the research of polymers in these directions. Therefore, the manuscript potentially deserves publication in Polymers, but the following points have to be carefully reconsidered before publication.
Answer: Thank you very much for your high evaluation.
- Please explain the conceptual difference between Self-Assembling and Self-Organizable.
Answer: Thank you very much for your suggestion. It has been provided on page 4 (Section 5). We have added the following lines: “We feel that at this time providing brief definitions of self-assembly and self-organization could be helpful. Self-assembly refers to the aggregation of a supramolecular object while self-organization refers to the process via which a periodic or quasiperiodic array generated from supramolecular objects is formed [70,101].” In addition, on page 36, in the conclusion section, we have added a short discussion as follows: “We also expect that this Perspective will impact on other fields of soft and living self-organized matter as it was already demonstrated by our laboratory for the case of helical self-organizations [262,263]. As elegantly pointed out by Lehn, there are unlimited fields of self-organization [264,265] where these concepts can and we hope will, provide an impact.”
- There is a problem with the formatting of lines 113 to 116, please revise it.
Answer: Thank you very much for pointing that. They are corrected (please see page 6).
- The format of “7.7 Covalent and Supramolecular Multiplicity of the Focal Point” is different from others, please revise it and check for the same type of problem.
Answer: Thank you very much for pointing that mistake. We have corrected it on page 16.
Reviewer 3 Report
Comments:
The stimuli-responsive principles of supramolecular organizations emerging from self-assembling and self-organizable dendrons, dendrimers and dendronized polymers is outlined for the first time in this invited Perspective. Therefore, the novelty of the manuscript is strong enough to be published in Polymers. The authors define the supramolecular organizations as stimuli or stimulus which originate from biology. Firstly, a historical story to the discovery and development of conventional and self-assembling and self-organizable dendrons, dendrimers and dendronized polymers is briefly depicted. Then a classification of stimuli-responsible principles as internal- and external-stimuli was made. This part is the focus in this Perspective. The former one is referred to variation or tuning of the primary and secondary structure of dendrons, dendrimers and dendronized polymers and the latter include environment, light, electric &magnetic field. Summarily, in this invited Perspective, the authors have interpreted and presented the relevant results correctly from the cited reference. Moreover, the manuscript is easy to understand and has a good accessibility for the researchers from the different research topic. I don’t have any scientific questions or doubts for this manuscript. Therefore, I recommend it to be accepted.
Author Response
The stimuli-responsive principles of supramolecular organizations emerging from self-assembling and self-organizable dendrons, dendrimers and dendronized polymers is outlined for the first time in this invited Perspective. Therefore, the novelty of the manuscript is strong enough to be published in Polymers. The authors define the supramolecular organizations as stimuli or stimulus which originate from biology. Firstly, a historical story to the discovery and development of conventional and self-assembling and self-organizable dendrons, dendrimers and dendronized polymers is briefly depicted. Then a classification of stimuli-responsible principles as internal- and external-stimuli was made. This part is the focus in this Perspective. The former one is referred to variation or tuning of the primary and secondary structure of dendrons, dendrimers and dendronized polymers and the latter include environment, light, electric & magnetic field. Summarily, in this invited Perspective, the authors have interpreted and presented the relevant results correctly from the cited reference. Moreover, the manuscript is easy to understand and has a good accessibility for the researchers from the different research topic. I don’t have any scientific questions or doubts for this manuscript. Therefore, I recommend it to be accepted.
Answer: Thank you very much for your high evaluation.